# Atmospheric carbon-dioxide variations across the middle Miocene climate transition

Markus Raitzsch[1,2], Jelle Bijma[2], Torsten Bickert[1], Michael Schulz[1], Ann Holbourn[3], Michal Kučera[1]

[1]MARUM – Zentrum für Marine Umweltwissenschaften, Universität Bremen, Leobener Straße 8, 28359 Bremen, Germany
[2]Alfred-Wegener-Institut, Helmholtz-Zentrum für Polar-und Meeresforschung, Am Handelshafen 12, 27570, Bremerhaven, Germany
[3]Christian-Albrechts-Universität, Institut für Geowissenschaften, 24118 Kiel, Germany

*Correspondence to*: Markus Raitzsch (mraitzsch@marum.de)

**Abstract.** The middle Miocene climate transition ~14 Ma marks a fundamental step towards the current "ice-house" climate, with a ~1 ‰ $\delta^{18}O$ increase and a ~1 ‰ transient $\delta^{13}C$ rise in the deep ocean, indicating rapid expansion of the East Antarctic Ice Sheet associated with a change in the operation of the global carbon cycle. The variation of atmospheric $CO_2$ across the carbon-cycle perturbation has been intensely debated as proxy records of $pCO_2$ for this time interval are sparse and partly contradictory. Using boron isotopes ($\delta^{11}B$) in planktonic foraminifers from ODP Site 1092 in the South Atlantic, we show that long-term $pCO_2$ varied at 402 k.y. periodicity between ~14.3 and 13.2 Ma and follows the global $\delta^{13}C$ variation remarkably well. This suggests a close link to precessional insolation forcing modulated by eccentricity, which governs the monsoon and hence weathering intensity, with enhanced weathering and decreasing $pCO_2$ at high eccentricity and vice versa. The ~50 k.y. lag of $\delta^{13}C$ and $pCO_2$ behind eccentricity in our records may be related to the slow response of weathering to orbital forcing. A $pCO_2$ drop of ~200 µatm before 13.9 Ma may have facilitated the inception of ice-sheet expansion on Antarctica, which accentuated monsoon-driven carbon cycle changes through a major sea-level fall, invigorated deep-water ventilation, and shelf-to-basin shift of carbonate burial. The temporary rise in $pCO_2$ following Antarctic glaciation would have acted as a negative feedback on the progressing glaciation and helping to stabilize the climate system on its way to the late Cenozoic "ice-house" world.

## 1 Introduction

With rapid cooling of Antarctica and associated expansion of the East Antarctic Ice Sheet (EAIS), the middle Miocene climate transition (MMCT) ~14 Ma marks a fundamental step towards the current "ice-house" climate (Woodruff and Savin, 1991; Flower and Kennett, 1994). The transition is characterized by a $\delta^{18}O$ increase of ~1 ‰ in the deep ocean between 13.9 and 13.8 Ma and a positive (~1 ‰) $\delta^{13}C$ excursion between 13.9 and 13.5 Ma indicating a fundamental change in global carbon-cycle dynamics (Shevenell et al., 2004; Holbourn et al., 2005, 2007). This carbon-isotope maximum (CM) event CM6 is the last and most prominent of a suite of maxima within a long-lasting $\delta^{13}C$ excursion, the "Monterey Excursion",

spanning ~16.9 to ~13.5 Ma and apparently paced by long-term changes in the 400 k.y. eccentricity cycle of the Earth's orbit (Vincent and Berger, 1985; Holbourn et al., 2007). These $\delta^{13}C$ maxima hint at a major reorganization of the marine carbon cycle, and the temporal coincidence of CM6 with expansion of the Antarctic ice sheet indicates that the glaciation may have been caused by cooling due to reduced $CO_2$ radiative forcing (Foster et al., 2012; Greenop et al., 2014), and potentially reinforced by low seasonality (Holbourn et al., 2005) and high albedo over Antarctica (Shevenell et al., 2004).

The CM events within the Monterey Excursion have been traditionally considered as episodes of increased organic-carbon ($C_{org}$) burial resulting in $^{12}C$-depleted seawater, atmospheric $CO_2$ drawdown and associated cooling and buildup of continental ice (Vincent and Berger, 1985; Flower and Kennett, 1993, 1994; Badger et al., 2013). Another explanation for the CM events refers to the "missing sink" mechanism (Lear et al., 2004), where large areas of Antarctic silicate rocks are covered by ice caps, which reduce chemical weathering and, thus, limit their potential as a sink for atmospheric $CO_2$ (Pagani

et al., 1999; Lear et al., 2004; Shevenell et al., 2008). Despite the different impacts on atmospheric $CO_2$, both hypotheses result in increasing marine $\delta^{13}C$, either by removal of $^{12}C$-enriched organic carbon from the water column or by increasing the carbon isotopic fractionation during photosynthesis at higher aqueous $[CO_2]$.

    Sosdian et al. (2020) proposed that different mechanisms were responsible for the ~3 M.y. lasting Monterey carbon isotope excursion and the shorter ~400 k.y. periodic CM events. The Monterey excursion may have been caused by volcanic

outgassing from Columbia River Flood Basalts, resulting in a $pCO_2$ rise, global warming and rising sea level (e.g., Hodell and Woodruff, 1994). In contrast, the CM events were associated with global cooling and ice-sheet growth, increases in nutrient delivery, marine productivity and organic carbon burial (the classic Monterey hypothesis), which ultimately led to a drawdown of atmospheric $CO_2$ and a rise in $\delta^{13}C$ (Sosdian et al., 2020). An alternate hypothesis is that the high $\delta^{13}C$ of CM events could have been caused by increased monsoon-driven weathering and nutrient supply to the ocean in low latitudes,

resulting in enhanced $C_{org}$ burial and $pCO_2$ drawdown, but which was outweighed by the concomitant increase in shallow-water carbonate production that removed alkalinity and hence released $CO_2$ (Ma et al., 2011).

    However, CM6 is unique as it is the most prominent among the CM events and its onset coincided with expansion of the Antarctic ice sheet. Hence, the interpretation of this event is complicated by additional processes that came into play, including a 55-75 m sea-level fall (Lear et al., 2010) that might have initiated a shelf-to-basin shift in carbonate burial,

carbonate and pyrite weathering on exposed shelf areas (e.g., McKay et al., 2016; Ma et al., 2018, Kölling et al., 2019), and changes in ocean circulation, bottom-water ventilation and deep-water production (e.g., Shevenell et al., 2004; Holbourn et al., 2007, 2013, 2014; Kuhnert et al., 2009; Tian et al., 2009; Knorr and Lohmann, 2014), all of which might have influenced $\delta^{13}C$ and $pCO_2$. For instance, in a recent study it was suggested that a northward shift of frontal systems may have reduced Southern Ocean upwelling, resulting in increased carbon storage in the deep ocean and thus a drawdown of atmospheric $CO_2$

(Leutert et al., 2020). It was proposed that the $\delta^{13}C$ excursion of CM6 could be caused by increased weathering of $^{13}C$-enriched shelf carbonates exposed after the sea-level fall and a terrestrial carbon reservoir expansion (Ma et al., 2018). As a consequence, the enhanced shelf carbonate weathering and carbon storage on land as well as a more sluggish meridional

Pacific Ocean overturning circulation, due to reduced deep-water formation in the Southern Ocean, resulted in a drawdown of $pCO_2$ (Ma et al., 2018).

The nature of the carbon cycle perturbation could be better constrained if we knew the evolution of atmospheric $CO_2$ across the MMCT. Understanding the role of the carbon cycle in this cooling step is a key to assess Earth-system sensitivity to $CO_2$ forcing and the long-term stability of the Antarctic ice sheet under rising $CO_2$ concentrations. However, most proxy records for the history of $pCO_2$ across the MMCT are incomplete or at low resolution, thus prohibiting resolution of the CM events (Pagani et al., 1999; Kürschner et al., 2008; Foster et al., 2012; Ji et al., 2018; Sosdian et al., 2018; Super et al., 2018) and

making it difficult to identify the mechanisms responsible for this major step into the "ice-house" world. The only high-resolution $pCO_2$ reconstruction across CM6 is based on alkenone $\delta^{13}C$ data from a Miocene outcrop on Malta (Badger et al., 2013), but it does not reveal a significant change.

Therefore, to better understand atmospheric $CO_2$ evolution over the period of EAIS expansion, we generated a relatively high-resolution reconstruction based on $\delta^{11}B$ measurements of fossil planktonic foraminifers from the South Atlantic (Figs.

1, 2). The boron isotopic composition of biogenic carbonates is a reliable recorder of ambient pH, which in turn is closely linked to atmospheric $pCO_2$ in oligotrophic surface waters. Our record fully captures the carbon isotope excursions CM5b and CM6, which include the major expansion of the EAIS.

## 1 Material and Methods

### 1.1 Sampling strategy

This research used samples from Ocean Drilling Program (ODP) Site 1092, located at 46.41 S and 7.08 E (Fig. 1) in a water depth of 1973 m. The interval studied (~178-184 mcd, see Table S1) consists of nannofossil ooze with excellent carbonate preservation, as shown by SEM-imaged undissolved coccoliths, which are susceptible to corrosion (Fig. 2). The shown example is from ~13.8 Ma, where pH was low, but shell preservation is similarly good throughout the entire record. From the 250-315 μm size fraction, approximately 200 specimens (~2.5 mg) of the cold-water dwelling planktonic foraminifer *G.*

*bulloides* were picked for $\delta^{11}B$ analysis from 35 processed 10 $cm^3$ sediment samples. In addition, about 5 to 8 specimens of the benthic foraminifer *Cibicidoides wuellerstorfi* were picked from 14 intervals to reconstruct deep-sea pH (Table S1). Today, surface-water $pCO_2$ at Site 1092 is close to equilibrium with the atmosphere ($\Delta pCO_{2sea-air} < -30$ μatm (Takahashi et al., 1993, 2009)). Plankton tow data revealed that *G. bulloides* in the Atlantic sector of the Southern Ocean is predominantly found within the upper 100 m of the water column (Mortyn and Charles, 2003). Based on DIC (dissolved inorganic carbon)

and TA (total alkalinity) from seasonal TCO2+TALK (Goyet et al., 2000), and temperature and salinity from WOA13 data sets, modern gradients in pH and $pCO_2$ are small at this site within the upper 200 m of the water column. During the austral spring season, when the shell flux of *G. bulloides* is highest (Jonkers and Kučera, 2015; Raitzsch et al., 2018), the

differences in pH and $p$CO$_2$ within this depth range are -0.03 and 17 µatm, respectively. Hence, we expect *G. bulloides* to be a reliable recorder of subsurface pH and $p$CO$_2$.

## 1.2 Age model

The age model used in this study is adopted from Kuhnert et al. (2009), but was revised to bring it in line with the astronomically tuned stable isotope record of the reference IODP Site U1338 (Holbourn et al., 2014) (Fig. 3, Table S2). For direct comparison of our $p$CO$_2$ record with the boron-based reconstructions available from the literature, the age models of ODP 761 (Holbourn et al., 2004, Foster et al., 2012, Sosdian et al., 2018) and the Blue Clay Formation of Ras-il-Pellegrin on Malta (Abels et al., 2005; Badger et al., 2013) were also re-tuned to match the reference curve U1338 (Table S2). Although the GSSP of the Langhian-Serravallian boundary (Mi-3b) in the Malta section was not explicitly placed at 13.82 Ma as suggested by Abels et al. (2005), the boundary in the revised age model (13.81 Ma) is in very close agreement, supporting the validity of the re-tuned record. Based on these revised age models, the δ$^{18}$O and δ$^{13}$C profiles of Sites 1092, ODP 791 and the Blue Clay Formation generally show a good agreement with those of Site U1338 (Fig. 3).

## 1.3 Boron isotope analysis

Foraminifer shells were cleaned following the protocol of Barker et al. (2003). Boron isotope ratios were measured following Raitzsch et al. (2018). Briefly, the cleaned samples were dissolved in 60 µL of 1 N HNO$_3$ and micro-distilled on a hotplate to separate boron from the carbonate matrix. The microdistillation method has been proven to yield a B recovery of ~100 %, a low procedural blank, and accurate results, even at low B concentrations (Gaillardet et al., 2001; Wang et al., 2010; Misra et al., 2014; Raitzsch et al., 2018). The procedural blank contribution was 10-50 pg B, which equates to ~0.2-0.8 % of the total [B] in the microdistillation vial. The distillate containing only boron was diluted with 2 % HNO$_3$ and analyzed for isotopes in triplicate using a Nu Plasma II multi-collector ICPMS at AWI (Bremerhaven, Germany) that is equipped with a customized detector array of 16 Faraday cups and 6 secondary electron multipliers (SEM), also termed ion counters (IC). $^{11}$B and $^{10}$B were collected in IC5 and IC0, respectively, at a boron concentration of ~3 ppb. As three of the SEMs were later replaced by Daly detectors, $^{11}$B and $^{10}$B of *C. wuellerstorfi* samples were collected in D5 and D0, respectively, at a boron concentration of ~2 ppb. $^{11}$B/$^{10}$B was standardized against concentration-matched NBS 951 using the standard-sample-standard bracketing technique, and frequent analysis of control standard AE121 with an isotopic composition similar to that of foraminifers was monitored to ensure accuracy of measurement. Measurement uncertainties are reported as 2 standard deviations (2σ) derived from triplicate measurements or as ±0.30 ‰ (for SEM-analyzed samples) and ±0.25 ‰ (for Daly-analyzed samples) determined from the long-term reproducibility (2σ of per-session δ$^{11}$B averages) of the control standard, whichever is larger (see Table S1).

### 1.4 Calculation of carbonate chemistry

To reconstruct pH and $pCO_2$ for the middle Miocene from the boron isotopic composition of foraminiferal shells, more constraints and assumptions of boundary conditions need to be made, as the ocean chemistry was different from today. This includes the boron isotopic composition of seawater ($\delta^{11}B_{sw}$), which has a direct effect on that of foraminifer shells ($\delta^{11}B_{foram}$), but also on the $\delta^{11}B_{foram}/\delta^{11}B_{borate}$ relationship, which in turn may lead to different pH yields. Further, we need to consider the seawater concentrations of calcium [Ca] and magnesium ions [Mg], which affect both the ocean's buffering capacity and the Mg/Ca ratios in foraminifer shells used to reconstruct seawater temperatures. Ultimately, we need to constrain a second carbonate system parameter (besides pH) to determine absolute atmospheric $pCO_2$ values as accurate as possible.

### 1.4.1 $\delta^{11}B$ of seawater

The most critical parameter for correct conversion of $\delta^{11}B_{foram}$ to pH is $\delta^{11}B_{sw}$, which is fortunately well constrained for the middle Miocene. We used a $\delta^{11}B_{sw}$ value of 37.80 ‰, which is the mean value from different independent studies and in close agreement with each other to within 0.2 ‰ (2σ) (Pearson and Palmer, 2000; Foster et al., 2012; Raitzsch and Hönisch, 2013; Greenop et al., 2017), and also comparable to the 38.5 ‰ modeled by Lemarchand et al. (2000). This value is used for both the calculation of pH and the adjustment of the $\delta^{11}B_{foram}/\delta^{11}B_{borate}$ regression intercept, where the effect is larger the more the slope differs from a 1:1 relationship (Greenop et al., 2019). Nonetheless, to inspect the influence of $\delta^{11}B_{sw}$ on final reconstructed $pCO_2$ values, we carried out the calculations using lower and upper extremes in $\delta^{11}B_{sw}$ of 36.80 ‰ and 38.80 ‰, respectively. The results show that the extreme scenarios not only reveal differences in reconstructed $pCO_2$ of up to 320 µatm, but also in relative changes, due to the non-linearity of the $\delta^{11}B$-pH proxy (Fig. S1). However, we propose that a $\delta^{11}B_{sw}$ value of 37.80 ‰ is a reasonable estimate, given the close agreement between independent studies and the closeness to non-boron based $pCO_2$ reconstructions.

### 1.4.2 Equilibrium constants

As the Miocene ocean had different [Ca] and [Mg] than today, their separate effects on seawater buffering must be taken into account to more accurately estimate past carbonate chemistry. For our study, all carbonate system parameters were calculated based on the MyAMI model (Hain et al., 2015; 2018) using a [Ca] and [Mg] of 13 mol/kg and 42 mol/kg, respectively, derived from halite fluid inclusions (Horita et al., 2002, Lowenstein et al., 2003; Timofeeff et al., 2006; Brennan et al., 2013).

### 1.4.3 Foraminiferal $\delta^{11}B$ calibrations

The different species-specific $\delta^{11}B_{foram}/\delta^{11}B_{borate}$ calibrations used to reconstruct Miocene pH may ultimately result in significantly different $pCO_2$ estimates (Fig. S2). For *G. bulloides*, there are three calibrations available (Martínez-Botí et al.,

2015; Henehan et al., 2016; Raitzsch et al., 2018), all of which yield very similar results (Fig. S2A). In this study, we used the one from Raitzsch et al. (2018) giving values that lie between the other two estimates. By contrast, for *Trilobatus trilobus* $\delta^{11}B$ the *Trilobatus sacculifer* calibration is applied, with four available equations (Foster et al., 2012; Martínez-Botí et al., 2015; Henehan et al., 2016; Dyez et al., 2018). The calibrations refer to different size fractions of *T. sacculifer* and different analytical techniques, resulting in Miocene $pCO_2$ estimates that differ from each other by up to ~2000 µatm (Figs. S2B,C). The very high $pCO_2$ (low pH) values obtained with the equation from Dyez et al. (2018) are attributed to the much shallower slope (data collected with N-TIMS), compared to those of the other calibrations (based on MC-ICPMS measurements). Due to the lower $\delta^{11}B_{sw}$ in the Miocene, this lower sensitivity also requires a stronger adjustment of the regression towards a higher intercept (Greenop et al., 2019), which results in calculations of even lower pH values. Hence, we assume that the calibration of Dyez et al. (2018) is not suitable for deep-time studies, when the $\delta^{11}B_{sw}$ was much different from today, using MC-ICPMS. However, the calibrations of Martínez-Botí et al. (2015) and Henehan et al. (2016) also result in Miocene $pCO_2$ estimates up to 400 µatm higher than the *G. bulloides* data (Fig. S2A) and other, non-boron-based, $pCO_2$ proxy data. Therefore, we retained the original equation provided by Foster et al. (2012) to re-evaluate the $\delta^{11}B$ data of their and the study by Badger et al. (2013).

### 1.4.4 pH and temperature estimates

$\delta^{11}B_{foram}$ was converted to $\delta^{11}B_{borate}$ using the calibration for *G. bulloides* from Raitzsch et al. (2018), which was adjusted for the lower $\delta^{11}B_{sw}$ following Greenop et al. (2019). Subsequently, pH was calculated using sea-surface temperatures (SST) derived from foraminiferal Mg/Ca ratios, a hydrostatic pressure of 5 bar (equivalent to 50 m water depth), and salinity based on the study by Kuhnert et al. (2009). The latter was estimated by converting $\delta^{18}O_{sw}$, derived from planktonic foraminiferal $\delta^{18}O$ and Mg/Ca temperatures (Shackleton, 1974), to salinity using a $\delta^{18}O_{sw}$:salinity gradient of 1.1 ‰ (the change in $\delta^{18}O_{sw}$ per salinity unit). This minimum gradient is required to keep the upper ocean density difference across the Subantarctic Front to enable the formation of Antarctic Intermediate Water (Kuhnert et al., 2009), which existed as Southern Component Intermediate Water (SCIW) since at least ~16 Ma (Shevenell and Kennett, 2004) and spread into all oceans adjacent to the Southern Ocean (Wright et al., 1992). The relative salinity record was then adjusted by 17.2 to achieve post-glaciation values similar to today. The reconstructed relative salinity change across the MMCT at Site 1092 is slightly more than 1, which is equivalent to the salinity gradient across the Subantarctic front today.

As seawater pH has a profound effect on foraminifer shell Mg/Ca, biasing reconstructed SSTs, which in turn results in altered pH estimates, we followed the method of Gray and Evans (2019) to iteratively solve Mg/Ca temperatures and $\delta^{11}B$-derived pH. For this, we modified the 'MgCaRB' R script of Gray and Evans (2019) to allow salinity and $\delta^{11}B_{sw}$ as additional input parameters (see code S1 in the supplement). In addition, we implemented a correction of shell Mg/Ca for changes in seawater Mg/Ca ($[Mg/Ca]_{sw}$) using the method of Evans and Müller (2012), where we applied the power function constants

for *Trilobatus sacculifer*, assuming that they are valid for other low-Mg foraminifers, too. In this study, we used a Mg/Ca$_{sw}$ that was likely in the order of 3.2 mol/mol at ~14 Ma ago, inferred from the chemical composition of halite fluid inclusions (Horita et al., 2002, Lowenstein et al., 2003; Timofeeff et al., 2006; Brennan et al., 2013).

To better compare our record with published boron-based *p*CO$_2$ reconstructions, the δ$^{11}$B and Mg/Ca data of *T. trilobus* from Foster et al. (2012) and Badger et al. (2013) were re-evaluated using the same calculation procedure as for our data. Here, we used the calibration for *T. sacculifer* from MgCaRB, although there is no discernable pH effect on the Mg/Ca of this species. Further, we applied a constant salinity of 35 and a hydrostatic pressure of 0 bar, but their effects on the final results are almost negligible as well.

Deep-sea temperatures were derived from Mg/Ca of *C. wuellerstorfi* (data courtesy of H. Kuhnert) using the species-specific calibration of Raitzsch et al. (2008), but we multiplied the pre-exponential constant with 0.825 (Evans and Müller, 2012) to correct temperatures for the lower Mg/Ca$_{sw}$ of the Miocene (see caption Table S1). Deep-ocean pH from the same species was calculated using a δ$^{11}$B$_{foram}$/δ$^{11}$B$_{borate}$ calibration (Table S1) fitted through core-top data from Rae et al. (2011) and Raitzsch et al. (2020), a salinity of 34, and a paleo-water depth of 1794 m.

The uncertainties for all pH and temperature values reported here were propagated using Monte Carlo simulations (10,000 repetitions) and comprise quoted 2σ uncertainties for δ$^{11}$B and Mg/Ca measurements, δ$^{11}$B$_{foram}$/δ$^{11}$B$_{borate}$ calibrations, δ$^{11}$B$_{sw}$ (±0.2 ‰), salinity (±1), and for the reciprocally corrected pH and temperatures calculated by MgCaRB (Gray and Evans, 2019).

### 1.4.5 *p*CO$_2$ reconstruction

To calculate *p*CO$_2$ from sea-surface pH, a second carbonate system parameter needs to be constrained. For this study, we used a sea-surface TA of 2000 µmol/kg, which is in line with various carbon cycle model results (Tyrrell and Zeebe, 2004; Ridgwell, 2005; Caves et al., 2016; Sosdian et al., 2018, Zeebe and Tyrrell, 2019), and we applied it to the entire record. However, to verify the validity of our approach we tested the effect of varying TA on *p*CO$_2$ calculations (see section 3.3). Uncertainties of *p*CO$_2$ estimates were fully propagated from adjusted 2σ uncertainties in pH and temperature (produced by

'MgCaRB'; see section 1.4.4), TA (±150 µmol/kg), and salinity (±1 psu) using the 'seacarb' package (Lavigne et al., 2011) programmed in 'R'. The applied temperature uncertainty is similar to the mean difference of ~2 °C between the non-corrected and pH-corrected Mg/Ca temperatures (Fig. S3), while the TA uncertainty encompasses the mean standard deviation of ±~130 µmol/kg between different TA models and the standard deviation of ±~50 µmol/kg for each model across the MMCT (Sosdian et al., 2018). The applied salinity uncertainty encompasses the potential salinity change across the

MMCT.

## 3. Results

### 3.1 Carbon dioxide record

Our boron-based record indicates a steady $p$CO$_2$ increase from ~380 to 520 ± 100 µatm between 14.25 and 14.08 Ma and a subsequent decrease to ~320 µatm until 13.87 Ma, culminating after the onset of the major ice-sheet expansion (Fig. 4E). It should be noted that this ~200 µatm drop in $p$CO$_2$ occurred at a time of northward shifting Southern Ocean fronts, which is only related to a step-like drop in SST and salinity around 14 Ma at this site (Fig. 4C) (Kuhnert et al., 2009). At 13.82 Ma, when SST had already reached a lower stable level and after the inception of the ~1 ‰ rise in benthic $\delta^{18}$O (Fig. 4A), $p$CO$_2$ increased rapidly by ~100 µatm and displayed high-amplitude variations of more than 50 µatm until 13.57 Ma. This transient rise in $p$CO$_2$ ended with a decrease to ~320 µatm and much reduced variability after 13.53 Ma (Fig. 4E). Despite the high 2$\sigma$ uncertainty of ~100 µatm for the overall $p$CO$_2$ record, the interpretation of comparatively small changes of ~50 µatm is reasonable, due to the smaller uncertainty of relative variations. The latter is estimated to be approximately 50 µatm (2$\sigma$), that is, when uncertainties in the $\delta^{11}$B$_{foram}$/$\delta^{11}$B$_{borate}$ calibration and $\delta^{11}$B$_{sw}$ are neglected. Sea-surface pH values within the entire record range from 7.94 to 8.11 ± 0.09, while deep-sea pH varies between 7.77 and 7.87 ± 0.08 over the same interval (Figs. 4D and S4). Interestingly, the pH offset between the surface and deep ocean is nearly identical to the modern gradient and apparently does not change substantially within our Miocene record (Fig. S4), but the high uncertainties associated with the pH estimates hamper a reasonable statistical analysis of the temporal change in gradient.

The $p$CO$_2$ variations within the studied time interval show a remarkable agreement with variations in $\delta^{13}$C that correspond to CM5b and CM6 (Fig. 4). Accordingly, the maxima at ~14.1 and 13.7 Ma as well as the minima at ~13.9 and 13.5 Ma indicate that atmospheric CO$_2$ levels were paced by 400 k.y. cycles, as also demonstrated by evolutive harmonic and power spectral analyses (Fig. 5), but due to the phase lag it is difficult to substantiate a direct link between long eccentricity cycles and atmospheric CO$_2$.

### 3.2 Sea-surface temperatures

Due to the profound effect of pH on Mg/Ca in shells of *G. bulloides*, which might bias reconstructed SSTs, we applied the modified MgCaRB tool of Gray and Evans (2019) to iteratively solve Mg/Ca temperatures and $\delta^{11}$B-derived pH. In addition, we applied a correction factor to account for the effect of [Mg/Ca]$_{sw}$ on shell Mg/Ca using the method of Evans and Müller (2012). However, to evaluate the difference between iteratively adjusted and non-adjusted pH and temperature values, we also performed the conventional calculations using the *G. bulloides* $\delta^{11}$B calibration from Raitzsch et al. (2018) and Mg/Ca-to-temperature component from Gray and Evans (2019), respectively. It should be noted that the Mg/Ca sensitivity of *G. bulloides* to temperature determined by Gray and Evans (2019) is much larger than that by Mashiotta et al. (1999). Therefore, the calculated unadjusted temperature decrease between 14.05 and 13.85 Ma is in the order of 13 °C, compared to ~6 °C determined with the equation of Mashiotta et al. (1999) and reported in Kuhnert et al. (2009). After the iterative

correction of Mg/Ca and pH, the results show that before the EAIS expansion the adjusted temperature and $pCO_2$ values are up to ~3 °C and ~60 µatm lower, respectively (Fig. S3). On the contrary, after 13.9 Ma the adjusted temperature and $pCO_2$ reconstructions are very similar, suggesting that approximately one fourth of the observed pre-glacial Mg/Ca decrease was induced by a contemporaneous pH rise.

### 3.3 Sensitivity tests

Given that we use TA as the second variable to calculate the inorganic carbon chemistry, which might affect calculated $pCO_2$ discernibly, we carried out a number of sensitivity tests. The first tests demonstrate that either constant salinity, a ~1 unit change in salinity, or TA co-varying with salinity using a modern TA-S relationship (~42 µmol/kg per S unit), has no discernible effect on $pCO_2$ estimates (Fig. S5A). As ODP Site 1092 was apparently influenced by a northward migration of the Southern Ocean fronts, we tested the potential impact on calculated $pCO_2$ by coupling TA to temperature. Even at an unrealistically high total change of 400 µmol/kg, calculated relative $pCO_2$ changes are similar after the EAIS expansion at ~13.9 Ma (Fig. S5B). Given the modern sea-surface TA gradient across the polar frontal system of ~100 µmol/kg (higher in the south), we conclude that the increasing influence of southern-sourced surface waters on $pCO_2$ estimates at our study site is comparatively small. When in our simulations TA was coupled to pH (Figs. S5C,D) or eccentricity (Figs. S5E-H) by ±100 µmol/kg, $pCO_2$ reconstructions are almost identical among each other. On the other hand, at TA variations of ±400 µmol/kg, the $pCO_2$ estimates may differ substantially between the different scenarios, but the general shape of the record remains more or less the same. Given that total variations in TA of 800 µmol/kg are very unlikely, we suggest that potential variations in TA do not affect our $pCO_2$ estimates significantly.

### 4. Discussion

### 4.1 Origin of Site 1092 $pCO_2$ signal

At face value, the reconstructed $pCO_2$ curve displays a remarkable resemblance to the global benthic $\delta^{13}C$ record (Fig. 4), suggesting that there is a first order link between the two. Our record, however, is the first that shows an increase of $pCO_2$ after the onset of Antarctic glaciation, which is counterintuitive at first glance, given that most hypotheses refer to enhanced $C_{org}$ burial as the main mechanism having caused the rise in $\delta^{13}C$ and presumably an atmospheric $CO_2$ drawdown. Hence this raises the question as to whether Site 1092 truly records a global $pCO_2$ signal, or rather a local one. It is difficult to tackle this question, because we neither have a complete, independent $pCO_2$ reconstruction for this time interval at similar or higher resolution for comparison, nor can we ascertain that any site millions of years ago was not influenced by local processes affecting the geochemical signal. Therefore, we cannot completely rule out a local or regional imprint on our $pCO_2$ record, particularly due to the proximity of Site 1092 to the polar front (Fig. 1).

There is indication for development of the subantarctic ocean frontal system and a better stratification of the upper ocean at Site 1092 following Antarctic ice-sheet expansion, suggested by a divergence in $\delta^{18}O$ between shallow- and deep-dwelling planktonic foraminifera (Paulsen, 2005). Although the occurrence of ocean fronts should have involved changes in biological productivity that may result in altered surface-water $[CO_2]$, the opal and $C_{org}$ contents in the sediment remain

steadily low throughout the middle Miocene (Diekmann et al., 2003), suggesting that Site 1092 was in an oligotrophic setting within the time interval studied. Moreover, if the $\delta^{11}B$ signal was mainly driven by a regional change in stratification after 13.85 Ma, this would not explain the low pH earlier during CM5, as shown in our record (Fig. 4).

Further, a better stratification should have resulted in a reduced vertical mixing of the water column and thus to reduced sea-air gas exchange, leading to decreased atmospheric $CO_2$, such as in the Southern Ocean during the Last Glacial Maximum

(François et al., 1997). On the contrary, our data do not imply such a situation, as the pH gradient between deep and surface ocean appears to have remained constant (Fig. S4), suggesting that DIC accumulated in both shallow and deep waters at this site, which resulted in an overall pH decrease. Indeed, Leutert et al. (2020) suggested that the expansion of the Antarctic Circumpolar Current (ACC) following ice-sheet expansion could have moved the frontal systems and westerly winds further north, which reduced Southern Ocean upwelling and hence provided a mechanism to increase carbon storage in the deep

ocean, drawing down $p$CO$_2$. If the deep ocean in the south was indeed less well-ventilated, we would expect benthic $\delta^{13}C$ to decrease after the onset of the EAIS expansion, because the ocean had more time to accumulate $^{13}C$-depleted $CO_2$ from decomposed $C_{org}$, which is the opposite (increasing $\delta^{13}C$, Fig. 4). All these complications call for more proxy records at sufficient resolution from spatially distinct locations to better constrain $p$CO$_2$ evolution across the MMCT.

### 4.2 Comparison with other $p$CO$_2$ records

Our $p$CO$_2$ data broadly agree with reconstructed long-term trends based on paleosol $\delta^{13}C$ (Ji et al., 2018) and planktonic foraminiferal $\delta^{11}B$ (Foster et al., 2012), except for the higher value at ~14.2 Ma (Fig. 6). The boron isotope data from the Mediterranean Sea by Badger et al. (2013) also show $p$CO$_2$ values that match our data well after 13.8 Ma, but our data do not exhibit a generally higher $p$CO$_2$ level immediately before 13.8 Ma. On the one hand, the relative pH increase of ~0.1 accompanying Antarctic glaciation was backed up by comparing a pH-dependent (Mg/Ca) with independent temperature

proxies (clumped isotopes and TEX$_{86}$) obtained from the same set of samples (Leutert et al., 2020). On the other hand, the strong SST decrease of ~4-5 °C after the onset of Antarctic glaciation, as suggested by the Mediterranean data, is more than 2 °C larger than those recorded in other low-latitude sites (Foster et al., 2012; Leutert et al., 2020; Sosdian and Lear, 2020), yet the reason for this discrepancy needs to be examined. In contrast, the $p$CO$_2$ reconstruction based on alkenone $\delta^{13}C$ data from Badger et al. (2013) is similar to the estimates of Super et al. (2018), which are generally lower than the boron-based

data and do not exhibit significant changes during CM6. Recent studies suggest that phytoplankton may enhance cellular

carbon supply for photosynthesis via carbon concentrating mechanisms under limited aqueous $[CO_2]$, making this proxy less sensitive at low to moderate $p$CO$_2$ levels (Badger et al., 2019; Stoll et al., 2019).

There is no other direct evidence for increasing pH immediately following the onset of EAIS expansion than the $\delta^{11}$B data by Badger et al. (2013), which is supported by Sosdian et al. (2020) who used planktonic foraminiferal B/Ca ratios from ODP
Site 761 (Fig. 1) to infer relative changes in DIC across the middle Miocene. As B/Ca in planktonic foraminifera is thought to be governed by the ratio between borate and DIC concentrations $[B(OH)_4^-/DIC]$, an increase in B/Ca can be interpreted as a decrease in DIC and/or an increase in $[B(OH)_4^-]$ (e.g., Allen et al., 2012). The high-resolution B/Ca record of Sosdian et al. (2020) reveals that the long-lasting Monterey excursion is accompanied by generally low B/Ca ratios, whereas CM intervals are characterized by higher B/Ca, suggesting periodic DIC decreases within a "high-DIC" world. The authors concluded that
during the CM events, ice sheet expansion and global cooling enhanced marine productivity and C$_{org}$ burial, which resulted in a reduction of surface-water DIC and $p$CO$_2$, and a rise in $\delta^{13}$C.

Within the overlapping period, which encompasses two full 400-k.y. cycles, our record exhibits some striking similarities with the B/Ca record of Sosdian et al. (2020), including the long-term decrease in $p$CO$_2$, accompanied by declining surface-water DIC, and the higher $p$CO$_2$ during CM5b (Fig. 6) that is accompanied by increased DIC. Intriguingly, the correlation
between the two records collapses at the onset of CM6, when the B/Ca record indicates a decline in DIC, which is corroborated by a lowering in $p$CO$_2$ (Badger et al., 2013). This is in contrast to our record showing that $p$CO$_2$ increased during CM6 (Fig. 4), although the records are once again in agreement at the end of CM6 (~13.5 Ma). This discrepancy could be attributed to non-identified factors influencing either of the two proxies, or regional effects on either of the two records. The potential influence of the polar front on Site 1092 is addressed in section 4.1, leading to the question whether
the data by Sosdian et al. (2020) could be alternatively explained by increased deep-water production. When the density of surface waters in the Southern Ocean increased due to cooling in a generally warm middle Miocene, the invigorated deep-ocean venting could have brought respired $CO_2$ to the atmosphere, leaving the overall ocean depleted in DIC and enriched in $CO_3^{2-}$, and possibly also in $\delta^{13}$C. The higher $CO_3^{2-}$ in turn could have favored the burial of carbonate in the deep sea, producing even more $CO_2$ that is vented to the atmosphere, as suggested for the Pleistocene glacial-interglacial cycles
(Toggweiler, 2008). This way, the loss of $CO_2$ in shallow waters, caused by enhanced marine productivity and C$_{org}$ burial (Sosdian et al., 2020), could have been outweighed by $CO_2$ added to the atmosphere by deep-ocean ventilation. However, this is speculative and needs to be validated in future studies.

It is worth noting that our $p$CO$_2$ record before Antarctic glaciation reveals a striking match with the box model output from Ma et al. (2011), while after EAIS expansion our values are some 50 µatm lower, although they still exhibit similar relative
changes (Fig. 6). The divergence after ~13.9 Ma may be related to the model by Ma et al. (2011), which was designed to simulate orbitally paced carbon reservoir changes for the Miocene Climate Optimum (MCO), and hence does not include ice sheets. Consequently, the model does not consider the effects of falling sea level on the shelf-to-basin fractionation of carbonate burial (McKay et al., 2016; Ma et al., 2018), or of ice-sheet expansion on ocean circulation and bottom-water

ventilation (e.g., Shevenell et al., 2004; Holbourn et al., 2007, 2013), all of which should have influenced $\delta^{13}C$ and $pCO_2$.

However, the similarity between our record and the model suggests that the ocean carbon reservoir varied in response to weathering-induced nutrient input, which in turn was controlled by varying orbital configuration. This model was sufficient to explain the observed $\delta^{13}C$ pattern of the MCO (Ma et al., 2011), but could also be capable of explaining a portion of the observed carbon cycle changes in a world with ice-covered Antarctica.

### 4.3 The role of weathering in the carbon cycle

The $pCO_2$ increase preceding and following the EAIS expansion closely tracks the global $\delta^{13}C$ history of CM5b and CM6 (Fig. 4), emphasizing a fundamental link between changes in the ocean carbon cycle and atmospheric carbon dioxide. The CM peaks correspond to minima in the 400 k.y. eccentricity cycle, suggesting that $\delta^{13}C$ variations were related to changes in monsoon and weathering intensity (Holbourn et al., 2007; Ma et al., 2011). Eccentricity maxima lead $\delta^{13}C$ minima at the 400 k.y. band by on average more than 50 k.y., which is comparable to the phase lag of $47 \pm 18$ k.y. observed in other Miocene

records, which can possibly be attributed to the slow response of weathering to orbital forcing (Holbourn et al., 2007). Monsoon intensity is paced by solar radiation variations caused by precessional cycles but their amplitude variations are modified by eccentricity. More precisely, higher eccentricity results in larger precession amplitudes and hence in larger wet/ dry variations in the tropics (e.g., Wang, 2009), stronger physical and chemical weathering, and ultimately in an increased input of particulate organic material, dissolved inorganic carbon, alkalinity and nutrients to the oceans (e.g., Clift and Plumb,

2008; Ma et al., 2011; Wan et al., 2009). The potential influence of alkalinity variations on our $pCO_2$ record is shown in Fig. S5 and evaluated in section 3.3.

The increased nutrient supply enhanced primary production and organic carbon burial, which in turn lowered $pCO_2$. At the same time, the burial of shallow and total calcium carbonate ($CaCO_3$) increased, while $CaCO_3$ burial in the deep ocean decreased (Holbourn et al., 2007). However, as the net burial of $CaCO_3$ (enriched in $^{13}C$) in relation to $C_{org}$ (depleted in $^{13}C$)

increased at high eccentricity, $\delta^{13}C_{DIC}$ values decreased, which is in line with the global isotope signature (Holbourn et al., 2007; Ma et al., 2011). Estimates of Miocene shallow-water carbonate burial and shelf-basin fractionation of carbonate accumulation are only available in rather low temporal resolution and do not capture orbital variability that would allow assessment of global trends (Boudreau and Luo, 2017; Boudreau et al., 2019; van der Ploeg et al., 2019). Another limitation is that attempts to estimate shallow water carbonate burial on orbital timescale are limited to regional studies in very

different environments (Kroon et al., 2000; Reuning et al., 2002; Williams et al., 2002; Aziz et al., 2003; Auer et al., 2015; Betzler et al., 2018; Ohneiser and Wilson, 2018; Reolid et al., 2019) that cannot be easily synthesized to provide global estimates.

An increase in carbonate burial in shallow seas would have removed alkalinity from seawater and thus lowered pH and released $CO_2$ to the atmosphere (e.g., Zeebe and Wolf-Gladrow, 2001) during 100 k.y. eccentricity maxima, however the

resolution of our record is too low to detect this. On longer timescales (400 k.y. cycles), increased alkalinity input from rivers, dissolution of deep-sea carbonates, and the enhanced burial of $C_{org}$ in tropical regions during eccentricity maxima might have contributed to the long-term decrease in $pCO_2$. This agrees well with the box model output from Ma et al. (2011), suggesting that during the long-eccentricity maxima $pCO_2$ decreased, when the monsoon was most intense and the organic carbon burial and river fluxes were high. Conversely, weathering and nutrient supply to the ocean in low latitudes decreased when eccentricity was low, resulting in a net decrease in the $CaCO_3$-to-$C_{org}$ burial ratio and hence in $CO_2$ release and higher $\delta^{13}C$.

The model of Ma et al. (2011) was designed to explain the long eccentricity signal in the $\delta^{13}C$ record throughout the warm MCO and hence does not consider processes caused by expanding Antarctic ice sheets and their impacts on the climate system. Indeed, the modeled $pCO_2$ reveals smaller amplitudes after ~13.9 Ma than our reconstruction (Fig. 6), and a less pronounced CM6 in relation to the previous CM events. Changes in shelf-basin partitioning of carbonate burial and enhanced global deep-water ventilation as well as increased weathering resulting from a major sea level fall at 13.8 Ma (Ma et al., 2018) are possible mechanisms to explain the higher amplitude of CM6. Additional mechanisms may have contributed to the amplification of CM6, including (1) increased $C_{org}$ burial by enhanced carbon sequestration through the biological pump linked to intensified upwelling in low latitude areas; (2) substantial outgassing of $CO_2$ from the deep ocean ($\delta^{13}C$ of approx. -1 to -2 ‰) into the atmosphere ($\delta^{13}C$ of approx. -6 to -6.5 ‰), which would not only result in increased atmospheric $pCO_2$ but also in a positive excursion in atmospheric $\delta^{13}C$. The size of the relevant reservoirs (38,000 Gt in the deep ocean vs. 750 Gt in the atmosphere, if the proportions were similar to today) would suggest that this process can be sustained over considerable time intervals; (3) the amplitude of the 405 k.y. cycle-paced $\delta^{13}C$ excursions depends on the size of the global marine DIC pool (e.g., Paillard and Donnadieu, 2014). This size may have decreased following the loss of shelf seas caused by the major regression associated with the onset of CM6, resulting in a higher amplitude $\delta^{13}C$ response to eccentricity forcing.

## 4.4 Enhanced glacial deep-water ventilation

As stated in the previous section, the $\delta^{13}C$ of CM6 might be related to global deep-water ventilation, due to cooling in the Southern Ocean that invigorated deep-water formation. The MMCT bears similarities to the Eocene-Oligocene transition (EOT) ~34 Ma ago, when Antarctic ice-expansion was also accompanied by a large positive $\delta^{13}C$ excursion, in terms of both magnitude and duration (Coxall et al., 2005). The $\delta^{13}C$ excursion following Antarctic glaciation was accompanied by a transient $pCO_2$ rise of more than 300 µatm (Pearson et al., 2009; Heureux and Rickaby, 2015), which bears similarity to the $pCO_2$ rise during the mid-Miocene EAIS expansion, although the amplitude was much larger. A modeling study suggested that a sea-level-fall-induced shelf-basin carbonate burial fractionation and a temporal enhancement of deep-water formation

(by 150 %) in the Southern Ocean were sufficient to produce both the positive $\delta^{13}C$ excursion and a transient $pCO_2$ rise by ~80 µatm in the EOT (McKay et al., 2016).

Similarly, ocean circulation may have intensified during the Miocene EAIS expansion, as the latitudinal temperature gradient increased and atmospheric circulation strengthened. This is supported by benthic foraminifers, Mn/Ca and XRF data from Southeast Pacific sites, which indicate improved deep-water ventilation and carbonate preservation after 13.9 Ma,

particularly during colder climate phases (Holbourn et al., 2013). Improved deep-water ventilation may also have led to enhanced advection of silica-rich waters toward low latitudes, culminating in an increased upwelling and diatom productivity between 14.04 and 13.96 Ma and between 13.84 and 13.76 Ma in the equatorial East Pacific, as indicated by massive opal accumulation found at IODP Site U1338 (Holbourn et al., 2014). We speculate that increased upwelling may have resulted in $CO_2$ outgassing, if the supply exceeded consumption by primary productivity.

Such a scenario involving Southern Ocean cooling that led to enhanced deep-water formation is also in line with the carbon-cycle box model by Toggweiler (2008), showing that the associated ocean venting could have transported respired $CO_2$ from deep waters to the atmosphere. Consequently, as the ocean was left depleted in DIC and enriched in $CO_3^{2-}$ (and possibly also in $\delta^{13}C$), this could have favored the burial of carbonate in the deep sea, removing TA and releasing even more $CO_2$ that is vented to the atmosphere (Toggweiler, 2008). The assumed rise in deep-ocean $CO_3^{2-}$ following EAIS expansion and

associated deepening of the carbonate compensation depth (CCD) is not contradictory to the benthic foraminiferal B/Ca ratios from the deep South China Sea that indicate an increase in the calcite saturation state spanning CM6, i.e. a temporary deepening of the CCD (Ma et al., 2018).

It is noteworthy that CM6 exhibits a more pronounced $\delta^{13}C$ change than all other CM events of the Monterey excursion, which is not reflected by our $pCO_2$ record but rather shows a similar amplitude of change as CM5b. One possible

explanation for this phenomenon is that enhanced deep-water formation during CM6 could have placed Site 1092 into a frontal system in the Southern Ocean, thus recording increasing aqueous $[CO_2]$ in a climate of decreasing atmospheric $pCO_2$, which might have also contributed to the divergence in the records of B/Ca (Sosdian et al., 2020) and $pCO_2$ from this study. However, based on our data we can only speculate on the unique nature of CM6 in comparison to other CM events, which must be further explored in future studies.

## 5. Conclusions


We conclude that $pCO_2$ variations across the MMCT were paced by 400 k.y. eccentricity cycles, where $pCO_2$ decreased at high eccentricity and rose when eccentricity was low (Fig. 4). At high eccentricity, the global monsoon and hence weathering intensity increased (e.g., Wang, 2009), causing an increased input of dissolved inorganic carbon, alkalinity and nutrients to the oceans. The resulting increase in the $CaCO_3$-to-$C_{org}$ burial ratio lowered $pCO_2$ and decreased $\delta^{13}C_{DIC}$ (Ma et

al., 2011), and both lag behind eccentricity by ~50 k.y., likely due to the slow response of weathering to climate forcing. A

re-evaluation of foraminiferal Mg/Ca data from Site 1092 (Kuhnert et al., 2009) by iterative correction between Mg/Ca and pH (Gray and Evans, 2019) revealed that the Mg/Ca decrease starting ~180 k.y. before Antarctic ice-sheet expansion is too large to be mainly explained by a concomitant increase in pH. Our data rather suggest that at ~14.1 Ma sea-surface temperatures in the Atlantic sector of the Southern Ocean started to drop by ~10 °C, supporting the notion of Kuhnert et al.

(2009) that ocean fronts migrated northwards well before Antarctic glaciation. The concomitant $p$CO$_2$ decrease of ~200 µatm between 14.08 and 13.87 Ma might have crossed a threshold sufficiently to facilitate the inception of EAIS expansion at ~13.9 Ma. The associated sea-level fall could have accentuated the monsoon-driven carbon cycle changes through invigorated deep-water ventilation, shelf-to-basin partitioning of carbonate burial, and shelf weathering. The CO$_2$ rise after the onset of EAIS expansion, possibly caused by decreased weathering fluxes at low eccentricity and hence net decrease in

the CaCO$_3$-to-C$_{org}$ burial ratio and/or enhanced deep-ocean circulation, could have acted as a negative feedback on the progressing Antarctic glaciation. In this way, the radiative forcing due to the temporary $p$CO$_2$ rise may have helped to stabilize the climate system on its way to the late Cenozoic "ice-house" world. Our results highlight the need for more high-resolution $p$CO$_2$ records across the middle Miocene climate transition.

## Data availability

The boron isotope data collected for this study are available from Table S1, the tie points used for the revised age model are listed in Table S2, and the modified R code of 'MgCaRB' (Gray and Evans, 2019) is available from S1 in the supplement.

## Author contribution

MR and JB conceived the study (conceptualization). MR carried out measurements, analyzed the data, and performed data statistics (data curation, formal analysis, investigation). JB provided access to analytical instruments at AWI (resources). MR

raised funding for the project (funding acquisition). MR produced the figures for the manuscript (visualization), and wrote the first draft of the manuscript (writing – original draft). All authors interpreted, edited, and reviewed the manuscript (writing – review & editing).

## Competing interests

The authors declare that they have no conflict of interest.

## Acknowledgment

We thank the Ocean Drilling Program for providing samples from ODP Leg 177, and M. Maeke for washing the sediments and picking foraminifer shells. A. Benthien, B. Müller, K.-U. Richter, and U. Richter are thanked for their assistance in the lab. This study is part of a research project (RA 2068/3-1) granted to MR by the German Research Foundation (Deutsche Forschungsgemeinschaft DFG).

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

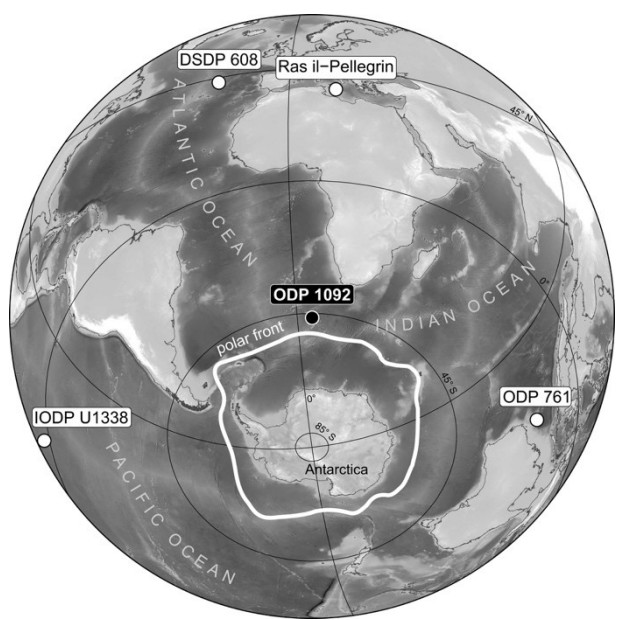

**Figure 1: Azimuthal view of the location of ODP Site 1092 (water depth 1973 m) on the Meteor Rise in the South Atlantic Ocean, north of the modern polar front. Also shown are locations of DSDP Site 608, ODP Site 761, IODP Site U1338 and Ras-il-Pellegrin on Malta, for which $p$CO$_2$ records are displayed in Fig. 6.**

710

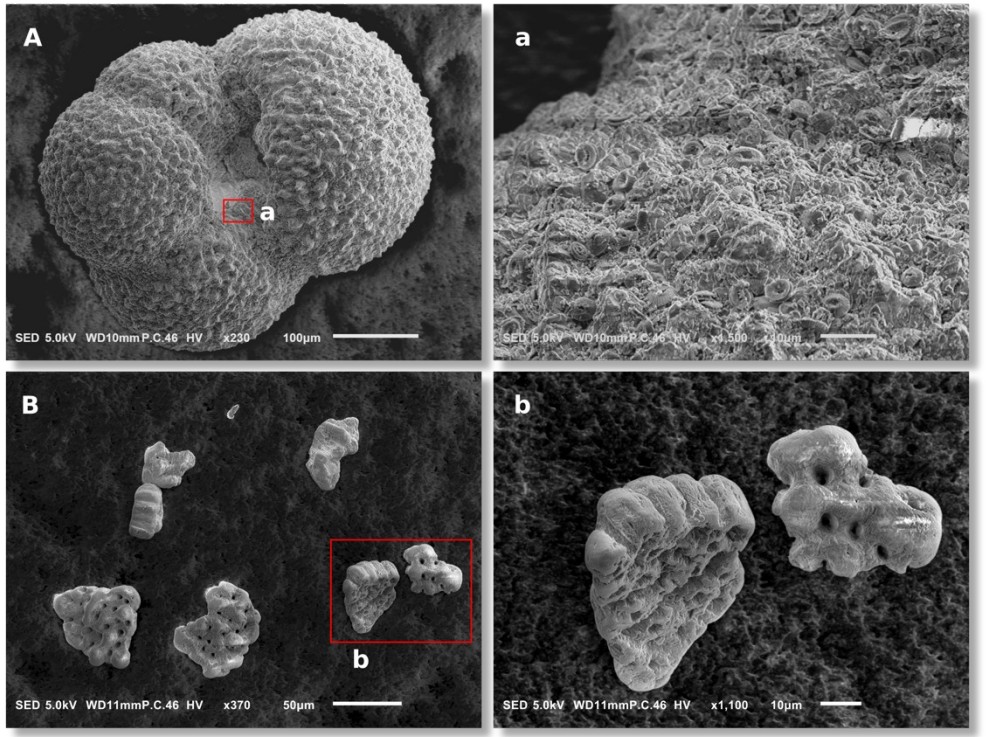

**Figure 2. SEM images of collected *G. bulloides* for δ¹¹B analysis (representative example from 177-1092B-18-4, 69-71 cm). (A) Whole shell; (a) close-up image reveal intact coccoliths covering the shell surface indicating lack of post-depositional dissolution. (B) Shell fragments after chemical cleaning; (b) close-up image shows absence of non-shell material.**

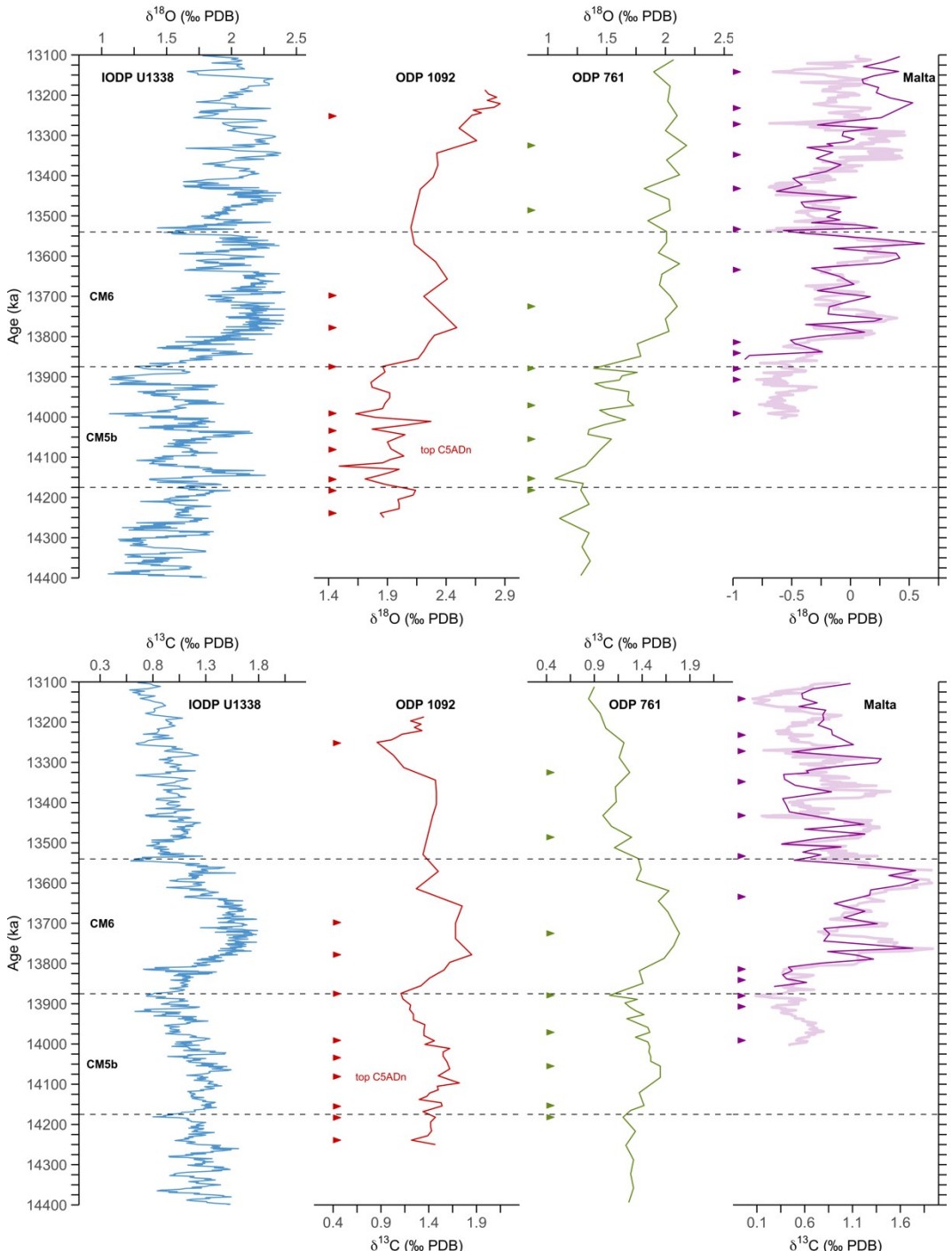

**Figure 3. Age models of ODP Site 1092, ODP Site 761 and Blue Clay Formation (Ras-il-Pellegrin, Malta), which were re-tuned with respect to the reference curve of IODP Site U1338 (Holbourn et al., 2014), based on their $\delta^{18}O$ (upper panel) and $\delta^{13}C$ (lower panel) records. Thick line of the Ras-il-Pellegrin record is from Abels et al. (2005), and thin line from Badger et al. (2013). Arrows mark tie points from this study, which are listed in Table S2. Dashed lines delimit CM events 5b and 6.**

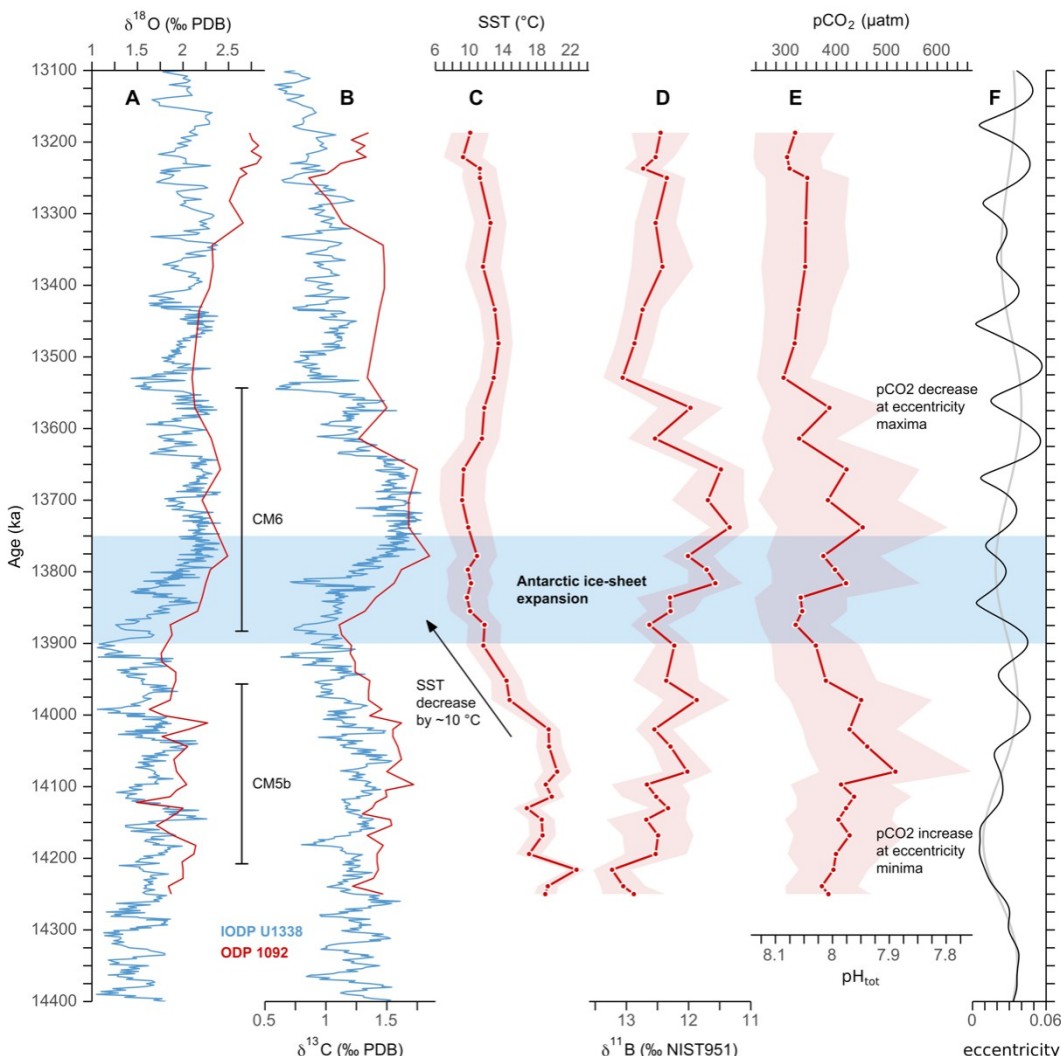

**Figure 4. Proxy records of the middle Miocene climate transition. (A) Benthic oxygen isotope curve from ODP Site 1092 (in red) used in this study and from eastern Pacific IODP Site U1338 (in blue) used as a reference (Holbourn et al., 2014). (B) The same as in A) but for carbon isotopes. The carbon maximum events CM5b and CM6 are indicated. (C) Site 1092 sea-surface temperatures based on *G. bulloides* Mg/Ca ratios (Kuhnert et al., 2009) and pH-adjusted following Gray and Evans (2012), with 2σ uncertainty band. (D) Raw boron isotope data of *G. bulloides*. Shaded area indicates 2σ uncertainties of measurement. (E) Estimated *p*CO₂ of surface waters basically using temperature-adjusted δ¹¹B-based pH following Gray and Evans (2019), a total alkalinity (TA) of 2000 ± 150 µmol/kg, and a δ¹¹B_sw value of 37.80 ± 0.2 ‰. Note that the secondary axis only approximates corresponding pH, and that shaded area delimits propagated uncertainties only for *p*CO₂. For exact pH values and uncertainties, see Fig. S4. (F) Eccentricity of the Earth's orbit from Laskar et al. (2004) and lowpass-filtered 400 k.y. signal (grey line).**

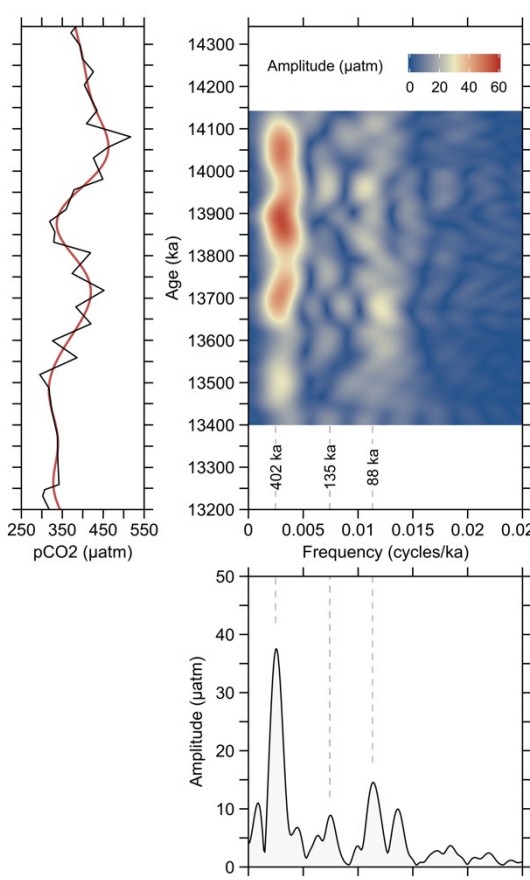

**Figure 5. Evolutive harmonic analysis (EHA) and power spectral analysis of *p*CO₂ record from ODP Site 1092, using the Thomson multitaper method (MTM). The red line in the left panel is the lowpass-filtered 400 k.y. signal using cosine-tapered window. Analyses were performed using the R package 'Astrochron' (Myers, 2014).**

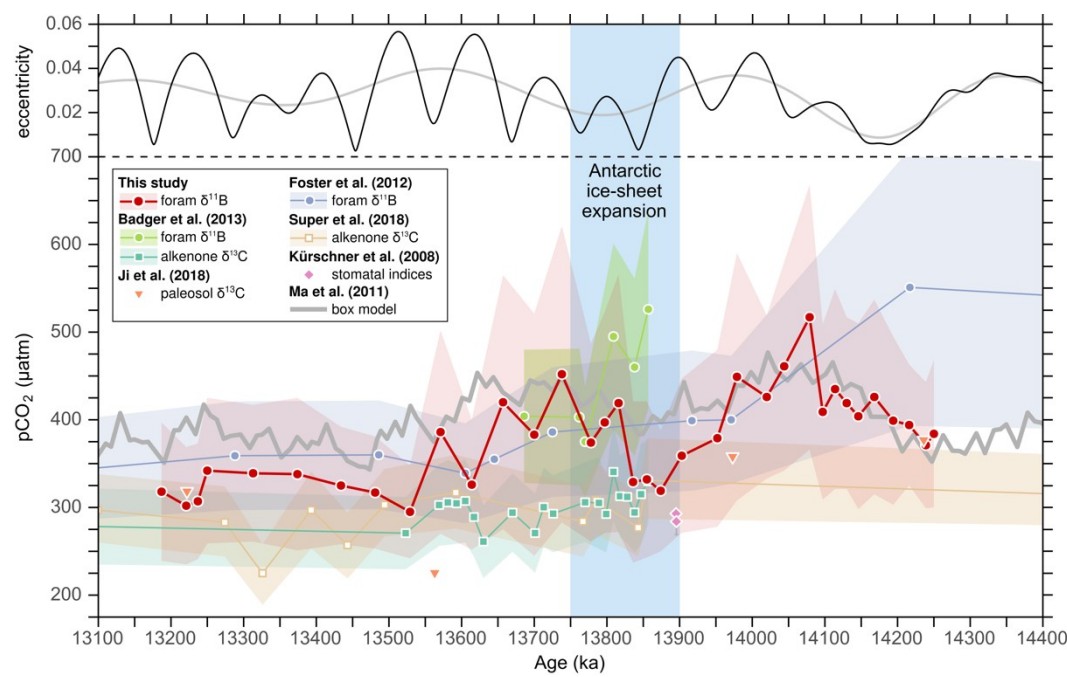

Figure 6. Eccentricity of the Earth's orbit from Laskar et al. (2004) (upper panel) and comparison of $p$CO$_2$ estimates from this study with literature data (lower panel). Note that for $p$CO$_2$ reconstructions from Foster et al. (2012) and Badger et al. (2013), the boron-based pH data were re-calculated using the MgCaRB tool (Gray and Evans, 2019) modified for deep-time reconstructions (code S1), a TA of 2000 ± 150 µmol/kg and a δ$^{11}$B$_{sw}$ value of 37.80 ± 0.2 ‰ for consistency with our data. In addition, the age models from these studies were also revised for direct comparison with our record (Fig. 3). All other records are based on their original data and age models.