# Peer review of "Atmospheric carbon-dioxide variations across the middle Miocene climate transition"

_Climate of the Past, 2020_

## Referee Comment (RC1) · Anonymous Referee #1 · 21 Aug 2020

Raitzsch et al present a new planktic boron isotope record across the mid Miocene Climate Transition, and interpret this in terms of changing CO2 and carbon cycle changes. The record enables deeper interrogation of potential climate and carbon cycle feedbacks during this important interval of global cooling. This interval also spans the end of the Monterey Excursion, which contains the well-known "carbon maxima" events. Having more CO2 data across this interval is very exciting, and thus the submission represents a substantial contribution worthy of publication in CP.

However, the narrative of the manuscript could be much improved – e.g. the conclusions are that a particular carbon cycle model is supported by the new records, but this model (or the mechanisms within it) are not referred to in the introduction. Much of the discussion hinges on assumptions of processes that have not been observed but are presented here as fact (e.g., that middle Miocene CO2 variations are caused by changes in shallow water carbonate production). The manuscript also fails to cite some key recent studies. However, I believe that if the narrative of the paper is improved and a few specific scientific questions are addressed then the robust interpretations of this fantastic dataset will be clearer and this will become a really exciting and useful contribution. Below I list the specific scientific questions I would like to see addressed, followed by more minor comments.

**1.	Is high eccentricity definitely associated with decreasing CO2? Would it be more robust to focus on the clear relationship between the d13C and CO2?**

The key finding presented in the abstract of the paper is that "long-term $p$CO2 variations between ~14.3 and 13.2 Ma were paced by 400 k.y. eccentricity cycles, with decreasing $p$CO2 at high eccentricity and vice versa."  I struggle to see this relationship in Figure 4 or Figure 5 – i.e. an inverse correlation between the pCO2 record and the thin grey line. I'm not saying the pCO2 record doesn't contain a long eccentricity signal, but the temporal relationship with the eccentricity curve could perhaps be demonstrated more robustly. While I struggle to discern a negative correlation between the authors' pCO2 record and the long eccentricity cycle, the relationship that does seem convincing is the positive relationship between d13C and CO2 across CM6 (Figure 4 and described in section 3.1). Perhaps therefore a clearer and simpler approach would be to set out different mechanisms in the introduction with their respective d13C-CO2 relationships. E.g., Holbourn et al 2007 ascribes the eccentricity signal in the longer d13C record to EITHER increased productivity and burial of Corg (the classic Monterey hypothesis) which would result in a negative correlation between d13C and CO2, OR a monsoon-driven increase in shallow water carbonate deposition, removing alkalinity from the oceans and releasing CO2, producing a positive correlation between d13C and CO2, as observed here. This explanation would be particularly useful because the increase in monsoon intensity variability has different impacts on the carbon cycle, but these are not discussed in the introduction. For example, an increase in nutrient delivery to the oceans and increasing Corg burial tends to increase d13C and decrease CO2. But in the Ma et al 2011 model this is outweighed by the concomitant increase in carbonate burial and associated CO2 release. It would be useful if this current manuscript could comment on the robustness of the relative importance of these processes in the model. However, see also my comment below about the uniqueness of CM6 - what might be expected to happen if for example the climate transition led to a general increase in marine productivity superimposed on these orbital

variations? Should we expect the d13C-CO2 relationship for CM6 to hold true for all CM events? This complication should be clearly addressed in the introduction also.

**2. What is the evidence for the increased shallow water carbonate deposition?**

As far as I know there is no direct evidence for eccentricity paced changes in shallower water carbonate burial. There are dissolution cycles in deep sea carbonates, but I am not aware of any study that rules out the influence of bottom water mass ventilation on these? I believe Holbourn et al 2007 favoured the monsoon hypothesis due to the 50kyr lag between eccentricity and d13C. But this lag is not discussed in this manuscript, and no supporting evidence is given for the statements that shallow water carbonate burial changed at this time.

**3.      Does the Site 1092 planktic d11B record global pCO2?**

This relationship between d13C and CO2 is key, and is independent of age model issues because the d13C has been recorded at each site and is a global signal. So it is interesting therefore that the Malta record shows a decrease in d11B-derived pCO2 associated with the onset of CM6 (increasing d13C) whereas the 1092 d11B-derived pCO2 shown here shows an increase. This raises the important question – is Site 1092 recording global atmospheric pCO2, or a more localised signal? A more local signal showing C storage in the high latitudes of the south Atlantic over the MMCT would also be very interesting of course. The change in the frontal positions is acknowledged in the text (Kuhnert et al 2009) but the change in stratification at this time is not mentioned. Paulsen (2005) (Bremen thesis) shows an increase in stratification at Site 1092 starting at 13.85Ma (using the divergence of surface:deep planktonic foraminiferal d18O). Could this stratification have been associated with an increase in surface water [CO2]? This possibility should be addressed in the manuscript.

**4.      Is it appropriate to compare the records across CM6 with those of a model that does not consider the impacts of the MMCT itself?**

CM6 immediately follows the ice growth of the MMCT and is the largest by far of the CM events, making its interpretation more complex than the other CM events. It follows a major sea level fall, which would have affected the shelf:basin burial of carbonate, d13C and CO2 (e.g., Mckay et al 2016, Ma et al 2018). Further, the ice advance was likely associated with changes in ocean circulation and ventilation of bottom water masses (with implications for both d13C and CO2). Is it therefore appropriate to make a straightforward comparison of the d13C and CO2 records to the Ma et al 2011 model without any consideration of these other processes? The Ma et al 2011 model was constructed to explain the long eccentricity signal in the d13C record throughout the Miocene Climatic Optimum, rather than examine CM6 specifically. It is solely forced by ETP, and does not include processes triggered by cryospheric thresholds in the climate system and resulting impacts. On the other hand, the Ma et al (2018) model suggests that a significant cause of the d13C increase at CM6 was the increased weathering resulting from the sea level fall. By saying the results here support the Ma et al 2011 model for CM6, where does that leave our understanding of the importance of shelf-basin carbonate deposition on global d13C signals?

**5.      How does the pCO2 record compare with the high-resolution B/Ca record of Sosdian et al 2020?**

It is odd that the manuscript does not compare the d11B-CO2 record with the high resolution B/Ca records published by Sosdian et al 2020. Those authors suggest that the increasing d13C of the CM events is associated with decreasing surface water DIC at Site 761. This is consistent with an increase in Corgburial:Carbonate burial which would predict a negative relationship between d13C and global CO2 across CM events, rather than the positive relationship observed across CM6 at Site 1092 here. Although, before a direct comparison can be made the origin of the d11B signal at Site 1092 needs to be thoroughly addressed.

**6.      What is the significance of the change in the Malta age model for the GSSP?**

The GSSP for the Langhian-Serravallian boundary is placed in the Malta section. There should be some comment about this in the age model section as the authors have changed the Malta age model.

**More minor comments:**
Line 45: *"However, most proxy records for the history of pCO2 across the MMCT are incomplete or at low resolution, thus prohibiting resolution of the CM events (Pagani et al., 1999; Kürschner et al., 2008; Foster et al., 2012; Ji et al., 2018; Sosdian et al., 2018; Super et al., 2018) and making it difficult to identify the mechanism responsible for the major step into the "icehouse" world."*
This statement is a bit disingenuous, these published records clearly show a significant pCO2 decrease just prior to the MMCT (see also Sosdian et al 2020). What is interesting of course is the higher resolution CO2 changes, and questions such as:  what caused the CM events? how do the CM events depend on background climate state? Why is CM6 (immediately following the major ice growth) the largest of the CM events?

Line 35: *"In a recent study, it was proposed that a more sluggish meridional Pacific Ocean overturning circulation, due to reduced deep-water formation in the Southern Ocean, enhanced the weathering of $^{13}$C-enriched shelf carbonates".*
This is incorrect, the enhanced weathering in this model was ascribed to the sea level fall. This model (Ma et al 2018) is very different to the Ma et al 2011 model. The Ma et al 2018 model treats the MMCT as an "event" whereas the Ma et al 2011 uses continually evolving orbital parameters as forcing. This important point is not explained in the current manuscript.

I find it odd that pH and pCO2 are plotted as the same curve with the same uncertainty envelope in Figure 4. pCO2 clearly has additional uncertainties (e.g. estimate of TA, d11Bsw), and I think these should be better represented in Figures 4 and 5.

Line 31: Need to also cite Foster et al 2012 as demonstrating climate-CO2 relationship across MMCT.

Careful with all references – e.g. Fig 3 caption "Badger et al 2015" should be "Badger et al 2013".
Line 168 – Fig 5 should be Fig 4.

---

## Referee Comment (RC2) · Anonymous Referee #2 · 7 Sep 2020

GENERAL COMMENTS Obtaining more B isotope-pH and CO2 estimates for the middle Miocene climate transition is a long overdue goal, making this study very timely and of great importance. The authors provide a comprehensive view of CO2 evolution for this period and potential mechanisms overlying potential eccentricity driven variations. The paper would benefit from some re-organization and focus on clarity, incorporation of recent studies in the discussion and the data, and a more comprehensive propagation of uncertainties, beyond the sensitivity analyses performed for alkalinity and salinity.

SPECIFIC COMMENTS Uncertainties: a)The analytical uncertainty reported is quite

small compared to the uncertainty of replicate analyses. There may be differences between the use of IC vs. Faraday detectors in different studies and here. Nevertheless, the authors should provide more details here as well on how they calculate analytical uncertainty. For example, the consistency standard used to calculate long term precision should have been run at similar concentrations to those of samples, and the uncertainty of this should be larger at low B-levels. Additionally, the authors should provide more details on the B blank contribution (if any). b)Why is d11Bsw error systematic? If weathering is extremely pronounced, couldn't this cause variations in d11Bsw across this time window, even if the average residence time for B may be longer? Even if so, because of the non-linearity of the d11B-pH proxy, at different d11Bsw the dpH and thus dCO2 could differ. Some could, thus, argue the uncertainty in d11Bsw encompasses both uncertainty in absolute value across the MMCT but also potential variations across the window. The authors should provide at minimum two scenarios based on minimum and maximum d11Bsw estimates for this period. c)The level of details in Fig. 7 with all sensitivity analyses is very much appreciated. However, could the authors provide more explanation on how they estimate the Alk and Temp uncertainties? If they compare to literature or proxy estimates, shouldn't they use the maximum uncertainty reported (i.e. $\pm$ 2C, and $\pm$ 130 umol/kg Alk)?

Comparing to other studies: The authors should discuss their results in light of two recent publications for the middle Miocene, Leutert et al. 2020 (Nat. Geo) for both their SST and dpH estimates, and Sosdian et al. 2020 (Nat. Comm.) for C cycle in relationship to climate.

Comparing CO2 records: e.g. Fig. 5: what drives the differences between different d11B records for the target age-window? Section 4.1 needs some more discussion, with focus on how this new record could differ from previous d11B records. Could there be an upwelling signal at the study site driving those high CO2 estimates when they deviate from the other records? It could also be differences in the calibration used for d11B, or the assumptions for calculating carbonate system parameters. It may be

wise to process the d11B records in the same manner, exclude the potential of any regional and variable CO2 disequilibrium, and then merge reliable d11B records into a single record with full propagation of uncertainties. If uncertainties are not propagated, and instead sensitivity runs are provided (i.e. d11Bsw), then better to display relative changes in pH/CO2 instead of absolute values.(Here it may be wise to remove the alkenone CO2 as they are not discussed enough beyond what is already available in the literature and thus do not contribute to the story.)

Focusing on the d11B records available and this new one, it would be also beneficial to display not only CO2 but also pH evolution across the MMCT, and how different records compare.

Site setting: It is argued that the site is not affected by upwelling being north of the frontal systems in the Southern Ocean. However, can this be said with certainty for the middle Miocene? Is there any evidence for that?

Carbonate system calculations: The authors should consider the effect of Mg and Ca concentrations in seawater on carbonate system calculations (i.e. K1, K2, Ksp), such as in Hain et al. 2015; 2018 or Zeebe and Tyrrell 2019.

Benthic-planktic pH records: Although the uncertainties are very large to make discernible conclusions about pH gradient values during the middle Miocene, it is interesting to further explore the dpH evolution and the surface-to-deep gradient evolution during the Miocene, and what drives this. If the benthic foraminiferal pH record is included, it should be discussed further.

Discussion on role of eccentricity and deep water ventilation: Here the section leaves us wanting more! It could benefit from some reorganization for clarity and flow, including recent studies such as those mentioned above.

---

## Author Comment (AC1) · 13 Oct 2020

Raitzsch et al present a new planktic boron isotope record across the mid Miocene Climate Transition, and interpret this in terms of changing CO2 and carbon cycle changes. The record enables deeper interrogation of potential climate and carbon cycle feedbacks during this important interval of global cooling. This interval also spans the end of the Monterey Excursion, which contains the well-known "carbon maxima" events. Having more CO2 data across this interval is very exciting, and thus the submission represents a substantial contribution worthy of publication in CP.

However, the narrative of the manuscript could be much improved – e.g. the conclusions are that a particular carbon cycle model is supported by the new records, but this model (or the mechanisms within it) are not referred to in the introduction. Much of the discussion hinges on assumptions of processes that have not been observed but are presented here as fact (e.g., that middle Miocene CO2 variations are caused by changes in shallow water carbonate production). The manuscript also fails to cite some key recent studies. However, I believe that if the narrative of the paper is improved and a few specific scientific questions are addressed then the robust interpretations of this fantastic dataset will be clearer and this will become a really exciting and useful contribution. Below I list the specific scientific questions I would like to see addressed, followed by more minor comments.

AC: We thank the referee for his constructive and thorough review of our manuscript. We will address all questions and comments in the following. The narrative of the manuscript will be changed during the revision.

1. Is high eccentricity definitely associated with decreasing CO2? Would it be more robust to focus on the clear relationship between the d13C and CO2? The key finding presented in the abstract of the paper is that "long-term pCO2 variations between ~14.3 and 13.2 Ma were paced by 400 k.y. eccentricity cycles, with decreasing pCO2 at high eccentricity and vice versa." I struggle to see this relationship in Figure 4 or Figure 5 – i.e. an inverse correlation between the pCO2 record and the thin grey line. I'm not saying the pCO2 record doesn't contain a long eccentricity signal, but the temporal relationship with the eccentricity curve could perhaps be demonstrated more robustly. While I struggle to discern a negative correlation between the authors' pCO2 record and the long eccentricity cycle, the relationship that does seem convincing is the positive relationship between d13C and CO2 across CM6 (Figure 4 and described in section 3.1). Perhaps therefore a clearer and simpler approach would be to set out different mechanisms in the introduction with their respective d13C-CO2 relationships. E.g., Holbourn et al 2007 ascribes the eccentricity signal in the longer d13C record to EITHER increased productivity and burial of Corg (the classic Monterey hypothesis)

which would result in a negative correlation between d13C and CO2, OR a monsoon-driven increase in shallow water carbonate deposition, removing alkalinity from the oceans and releasing CO2, producing a positive correlation between d13C and CO2, as observed here. This explanation would be particularly useful because the increase in monsoon intensity variability has different impacts on the carbon cycle, but these are not discussed in the introduction. For example, an increase in nutrient delivery to the oceans and increasing Corg burial tends to increase d13C and decrease CO2. But in the Ma et al 2011 model this is outweighed by the concomitant increase in carbonate burial and associated CO2 release. It would be useful if this current manuscript could comment on the robustness of the relative importance of these processes in the model. However, see also my comment below about the uniqueness of CM6 - what might be expected to happen if for example the climate transition led to a general increase in marine productivity superimposed on these orbital variations? Should we expect the d13C-CO2 relationship for CM6 to hold true for all CM events? This complication should be clearly addressed in the introduction also.

AC: The long response time of the carbon cycle appears to be consistent, when comparing other major d13C excursions (for instance, Cretaceous OAEs). However, it is difficult to determine phase relationships, when dealing with relatively short low-resolution records such as at Site 1092. In the highly resolved d13C U1338 record, there is a rebound during the onset of CM6, which corresponds to a precessional warming peak reflected in a transient d18O decrease. This feature is not captured at Site 1092. The age correlation between Sites U1338 and 1092 is in fact quite loose over the onset of CM6, making it difficult to evaluate the timing of the pCO2 increase in relation to the detailed evolution of d13C during the CM6 onset. Another issue is that the 400 kyr filtered curves that are often used to evaluate phase relationships between data sets can be strongly biased by the selected bandwidth. An important consideration here are asymmetries of the 405 kyr cycle in d13C that are not apparent in the filtered curves. Estimated phase relationships are based on the assumption that the corresponding signals are periodic. However, in case of the onset of the positive d13C excursions is

fast and would lead to artificial phase relationships when decomposed into sinusoidal base functions. This is especially the case for CM6. In contrast, the rates of change during the peak or plateau and recovery of the d13C excursion are much slower.

2. What is the evidence for the increased shallow water carbonate deposition? As far as I know there is no direct evidence for eccentricity paced changes in shallower water carbonate burial. There are dissolution cycles in deep sea carbonates, but I am not aware of any study that rules out the influence of bottom water mass ventilation on these? I believe Holbourn et al 2007 favoured the monsoon hypothesis due to the 50 kyr lag between eccentricity and d13C. But this lag is not discussed in this manuscript, and no supporting evidence is given for the statements that shallow water carbonate burial changed at this time.

AC: Estimates of Miocene shallow water carbonate burial and shelf-basin fractionation of carbonate accumulation are only available in rather low temporal resolution and do not capture orbital variability that would allow assessment of global trends. Some recent references for global trends include: • Boudreau & Luo, 2017, Earth and Planetary Science Letters, v. 474, p. 1–12, https://doi.org/10.1016/ j.epsl.2017.06.005. • Boudreau et al., 2019 Earth and Planetary Science Letters 512, 194–206. • Levitan, 2018 Fractionation of Carbonate Carbon (Đącarb) Accumulation between Continents and Oceans in the Late Mesozoic–Cenozoic. ISSN 0016-7029, Geochemistry International, 2018, Vol. 56, No. 7, pp. 702–710, Pleiades Publishing. Original Russian Text $^{©}$ M.A. Levitan, 2018, published in Geokhimiya, 2018, No. 7. • van der Ploeg, R. et al., 2019, Cenozoic carbonate burial along continental margins: Geology, v. 47, p. 1025–1028, https://doi.org/10.1130/G46418.1).

Another limitation is that attempts to estimate shallow water carbonate burial on orbital timescale are limited to regional studies in very different environments that cannot be easily synthesized to provide global estimates. The following regional studies have focused on: • Italy (Auer et al. (2015), https://doi.org/10.1002/2014PA002716), • Iberia (Abdul Aziz et al., 2003, Sedimentology, 50. pp. 211-236.

https://doi.org/10.1046/j.1365-3091.2003.00544.x.), • Bahamas (Kroon et al 2000, in Proceedings of the Ocean Drilling Program, Scientific Results, Vol. 166; Reuning et al 2002 Marine Geology 185 (2002) 121-142; Williams et al 2002, Marine Geology 185 (2002) 75-93), • Maldives (Betzler et al 2018 Prog Earth Planet Sci. 5.; Reolid et al 2019 Marine Geology 412 (2019) 199–216), • New Zealand (Ohneiser & Wilson 2018 Eccentricity-paced Southern Hemisphere glacial-interglacial cyclicity preceding the middle Miocene climatic transition. Paleoceanography and Paleoclimatology, 33, 795–806. https://doi.org/10.1029/2017PA003278).

We will add this information to the revised manuscript.

3. Does the Site 1092 planktic d11B record global pCO2? This relationship between d13C and CO2 is key, and is independent of age model issues because the d13C has been recorded at each site and is a global signal. So it is interesting therefore that the Malta record shows a decrease in d11B-derived pCO2 associated with the onset of CM6 (increasing d13C) whereas the 1092 d11B-derived pCO2 shown here shows an increase. This raises the important question – is Site 1092 recording global atmospheric pCO2, or a more localised signal? A more local signal showing C storage in the high latitudes of the south Atlantic over the MMCT would also be very interesting of course. The change in the frontal positions is acknowledged in the text (Kuhnert et al 2009) but the change in stratification at this time is not mentioned. Paulsen (2005) (Bremen thesis) shows an increase in stratification at Site 1092 starting at 13.85 Ma (using the divergence of surface:deep planktonic foraminiferal d18O). Could this stratification have been associated with an increase in surface water [CO2]? This possibility should be addressed in the manuscript.

AC: There is broad agreement that G. bulloides is a planktic foraminiferal species associated with upwelling of cold water at low latitudes. Accordingly, it is consistently used in the monsoonal upwelling zones of the Arabian Sea and equatorial Indian Ocean as an indicator of upwelling intensity. See classic papers of Kroon & Darling (1995) Journal of Foraminiferal Research, v. 25, no. 1, p. 39-52 and Peeters et al. (2002)

Global and Planetary Change 34 (2002) 269–291 and discussion of its utility as pCO2 indicator in upwelling regions by Palmer et al. (2010) Earth and Planetary Science Letters 295 (2010) 49–57. However, this is not the case at high latitudes, and we argue that the majority of G. bulloides lives in the upper 100 m within the mixed layer (Line 66). We will add in the revised text that the plankton-tow data we refer to are from the Atlantic sector of the Southern Ocean (Mortyn and Charles, 2003). Further, we will add the result by Diekmann et al. (2003) that the sedimentological record of this site hints at oligotrophic conditions throughout the Miocene interval. In addition, we think that if the boron isotope signal was mainly driven by a regional change in stratification after 13.85 Ma (Paulsen's data), as envisaged by the reviewer, this would not explain the low pH earlier during CM5, as shown in our record. We will briefly discuss the global/local origin of the signal in the revised version of the manuscript to address this criticism from both reviewers.

4. Is it appropriate to compare the records across CM6 with those of a model that does not consider the impacts of the MMCT itself? CM6 immediately follows the ice growth of the MMCT and is the largest by far of the CM events, making its interpretation more complex than the other CM events. It follows a major sea level fall, which would have affected the shelf:basin burial of carbonate, d13C and CO2 (e.g., Mckay et al 2016, Ma et al 2018). Further, the ice advance was likely associated with changes in ocean circulation and ventilation of bottom water masses (with implications for both d13C and CO2). Is it therefore appropriate to make a straightforward comparison of the d13C and CO2 records to the Ma et al 2011 model without any consideration of these other processes? The Ma et al 2011 model was constructed to explain the long eccentricity signal in the d13C record throughout the Miocene Climatic Optimum, rather than examine CM6 specifically. It is solely forced by ETP, and does not include processes triggered by cryospheric thresholds in the climate system and resulting impacts. On the other hand, the Ma et al (2018) model suggests that a significant cause of the d13C increase at CM6 was the increased weathering resulting from the sea level fall. By saying the results here support the Ma et al 2011 model for CM6, where does that leave our

understanding of the importance of shelf-basin carbonate deposition on global d13C signals?

AC: CM6 does exhibit major differences to the other 405 kyr cycles within the Monterey Excursion, as stated by the reviewer. This is a very interesting point and we will include a brief discussion of possible explanations for the difference in amplitude between CM6 and other Monterey CM events. Changes in shelf-basin partitioning of carbonate burial and enhanced global deep-water ventilation as well as increased weathering resulting from a major sea level fall at 13.8 Ma (Ma et al., 2018), as mentioned by the reviewer, are possible mechanisms to explain the higher amplitude of CM6. In addition, the following mechanisms may have contributed to the amplification of CM6: 1) Increased organic carbon burial by enhanced carbon sequestration through the biological pump linked to intensified upwelling in low latitude areas. 2) Substantial outgassing of CO2 from the deep ocean (d13C of approx. -1 to -2 ‰ into the atmosphere (d13C of approx. -6 to -6.5 ‰ would not only result in increased atmospheric pCO2 but also in a positive excursion in atmospheric d13C. The size of the relevant reservoirs (38000 Gt deep ocean vs. 750 Gt atmosphere, if the proportions were similar to today) would suggest that this process can be sustained over considerable time intervals. 3) The amplitude of 405 kyr cycle paced carbon isotope excursions depends on the size of the global marine DIC pool (e.g. Paillard & Donnadieu, 2014, Paleoceanography, A 100 Myr history of the carbon cycle based on the 400 kyr cycle in marine $\delta$13C benthic records https://doi.org/10.1002/2014PA002693). This size may have decreased following the loss of shelf seas caused by the major regression associated with the onset of CM6 – resulting in a higher amplitude d13C response to eccentricity forcing.

5. How does the pCO2 record compare with the high-resolution B/Ca record of Sosdian et al 2020? It is odd that the manuscript does not compare the d11B-CO2 record with the high resolution B/Ca records published by Sosdian et al 2020. Those authors suggest that the increasing d13C of the CM events is associated with decreasing surface water DIC at Site 761. This is consistent with an increase in Corgburial:Carbonate

burial which would predict a negative relationship between d13C and global CO2 across CM events, rather than the positive relationship observed across CM6 at Site 1092 here. Although, before a direct comparison can be made the origin of the d11B signal at Site 1092 needs to be thoroughly addressed.

AC: Although the new study by Sosdian et al. is an interesting and important study, we are not convinced that B/Ca in planktic foraminifera is a reliable proxy for borate/DIC or borate/bicarbonate ratios. While culture studies seem to clearly reveal such a relationship, our own experience showed that coretop and downcore samples are not that straightforward, and in some cases reveal opposite trends to what is expected from theoretical and culture studies. In contrast to benthic B/Ca and d11B, which evolved into reliable tools for reconstructing calcite saturation state and pH, respectively, planktic B/Ca is still far from being established. However, in the revised manuscript, we will mention the paper of Sosdian et al.

6. What is the significance of the change in the Malta age model for the GSSP? The GSSP for the Langhian-Serravallian boundary is placed in the Malta section. There should be some comment about this in the age model section as the authors have changed the Malta age model.

AC: The Langhian-Serravallian boundary at 13.82 Ma is still coincident with the revised age model, even if it was not used as a tie point, confirming the general agreement between the different age models. This will be mentioned in the revised text.

More minor comments:

Line 45: "However, most proxy records for the history of pCO2 across the MMCT are incomplete or at low resolution, thus prohibiting resolution of the CM events (Pagani et al., 1999; Kürschner et al., 2008; Foster et al., 2012; Ji et al., 2018; Sosdian et al., 2018; Super et al., 2018) and making it difficult to identify the mechanism responsible for the major step into the "icehouse" world." This statement is a bit disingenuous, these published records clearly show a significant pCO2 decrease just prior to the MMCT

(see also Sosdian et al 2020). What is interesting of course is the higher resolution CO2 changes, and questions such as: what caused the CM events? how do the CM events depend on background climate state? Why is CM6 (immediately following the major ice growth) the largest of the CM events?

AC: It is correct that existing records show that pCO2 was higher before EAIS expansion than afterwards, but our statement that none of them is sufficient to fully resolve the CM events is not incorrect. It was certainly not our intention to belittle these very important studies, which mostly aimed at reconstructing long-term changes, while our record just focusses on CM5b and CM6. This will be attenuated in the revised manuscript.

Line 35: "In a recent study, it was proposed that a more sluggish meridional Pacific Ocean overturning circulation, due to reduced deep-water formation in the Southern Ocean, enhanced the weathering of 13C-enriched shelf carbonates". This is incorrect, the enhanced weathering in this model was ascribed to the sea level fall. This model (Ma et al 2018) is very different to the Ma et al 2011 model. The Ma et al 2018 model treats the MMCT as an "event" whereas the Ma et al 2011 uses continually evolving orbital parameters as forcing. This important point is not explained in the current manuscript.

AC: Correct, this will be changed and discussed in more detail in the revised version.

I find it odd that pH and pCO2 are plotted as the same curve with the same uncertainty envelope in Figure 4. pCO2 clearly has additional uncertainties (e.g. estimate of TA, d11Bsw), and I think these should be better represented in Figures 4 and 5.

AC: The uncertainties in the pCO2 record are propagated uncertainties from boron isotope measurements (i.e. translated to pH uncertainty), TA, S and T, using the 'seacarb' package. What is missing, and here the reviewer is right, are the uncertainties from T, S and the d11B_borate/d11B_foram calibration propagated into the pH uncertainty itself. This will be done using a MonteCarlo approach. As pH has the largest influence on pCO2 estimation, the resulting pCO2 uncertainty will be considerably greater as

shown here. The uncertainty in d11Bsw is a systematic error, due to the long residence time of boron in seawater, and we think it will thus give a wrong impression of uncertainty in terms of relative pH (and pCO2) changes. However, since also referee 2 asked to account for it, we will either also propagate the d11Bsw uncertainty into the pH uncertainty, or will put our record into an envelope of extreme d11Bsw scenarios. In addition, we will also account for the effects of seawater [Ca] and [Mg] on the dissociation constants of carbonic and boric acid, following Hain et al. (2015). Further, in the revised manuscript, we will also account for the effect of d11Bsw on the d11B_borate/d11B_calcite calibration, but which will not yield much different results, due to the ~1:1 slope of the calibration of this species (Greenop et al., 2019).

Line 31: Need to also cite Foster et al 2012 as demonstrating climate-CO2 relationship across MMCT.

AC: OK, will be corrected.

Careful with all references – e.g. Fig 3 caption "Badger et al 2015" should be "Badger et al 2013".

AC: Thanks for spotting this, will be corrected.

Line 168 – Fig 5 should be Fig 4.

AC: Thanks, will be corrected.

---

## Author Comment (AC2) · 13 Oct 2020

GENERAL COMMENTS Obtaining more B isotope-pH and CO2 estimates for the middle Miocene climate transition is a long overdue goal, making this study very timely and of great importance. The authors provide a comprehensive view of CO2 evolution for this period and potential mechanisms overlying potential eccentricity driven variations. The paper would benefit from some re-organization and focus on clarity, incorporation of recent studies in the discussion and the data, and a more comprehensive propagation of uncertainties, beyond the sensitivity analyses performed for alkalinity and salinity.

[Figure]

AC: We thank the referee for his constructive and thorough review of our manuscript. We will address all questions and comments in the following.

SPECIFIC COMMENTS Uncertainties: a) The analytical uncertainty reported is quite small compared to the uncertainty of replicate analyses. There may be differences between the use of IC vs. Faraday detectors in different studies and here. Nevertheless, the authors should provide more details here as well on how they calculate analytical uncertainty. For example, the consistency standard used to calculate long term precision should have been run at similar concentrations to those of samples, and the uncertainty of this should be larger at low B-levels. Additionally, the authors should provide more details on the B blank contribution (if any).

AC: The reviewer is right, the long-term reproducibility is taken from runs with different [B], but not that much as the referee thinks, as most analyses were performed using ion counters or Daly detectors. We will check the "Daily performance" file to extract the values derived from measurements at ∼2-3 ppb. However, the average uncertainty from the sample replicate analyses is not so far from the long-term reproducibility of a consistency standard, given that all uncertainties for replicates <0.3 permil are replaced with the long-term value of 0.3 permil. If the actual numbers are taken into account, the average uncertainty of replicate measurements is ∼0.35 permil, which is very close to the long-term reproducibility. This small difference of 0.05 permil may be due to the fact that we run our sample rotationally, i.e. like ABC-ABC-ABC, while the consistency standards are run consecutively at the beginning and the end of a session, such as AAA-sample sequence-AAA. This could result in a slightly higher replicate uncertainty for the samples, as the measurements are temporally distinct, which might reflect slightly changing plasma conditions over the course of a day. The change in sample sequencing is certainly something worth looking at in the future. The information on lab blanks is in detail described in our 2018 EPSL paper, but we will briefly echo it in the revised manuscript.

b) Why is d11Bsw error systematic? If weathering is extremely pronounced, couldn't

this cause variations in d11Bsw across this time window, even if the average residence time for B may be longer? Even if so, because of the non-linearity of the d11B-pH proxy, at different d11Bsw the dpH and thus dCO2 could differ. Some could, thus, argue the uncertainty in d11Bsw encompasses both uncertainty in absolute value across the MMCT but also potential variations across the window. The authors should provide at minimum two scenarios based on minimum and maximum d11Bsw estimates for this period.

AC: The uncertainty in d11Bsw is systematic, due to the long residence time of boron in seawater, and we think it will thus give a wrong impression of uncertainty in terms of relative pH (and pCO2) changes. However, since also referee 1 asked to account for it, we will either also propagate the d11Bsw uncertainty into the pH uncertainty, or will put our record into an envelope of extreme d11Bsw scenarios, as suggested by the referee. In addition, in the revised manuscript, we will account for the effect of d11Bsw on the d11B_borate/d11B_calcite calibration, but which will not yield much different results, due to the ∼1:1 slope of the calibration for G. bulloides (Greenop et al., 2019).

c) The level of details in Fig. 7 with all sensitivity analyses is very much appreciated. However, could the authors provide more explanation on how they estimate the Alk and Temp uncertainties? If they compare to literature or proxy estimates, shouldn't they use the maximum uncertainty reported (i.e. ± 2C, and ± 130 umol/kg Alk)? Comparing to other studies: The authors should discuss their results in light of two recent publications for the middle Miocene, Leutert et al. 2020 (Nat. Geo) for both their SST and dpH estimates, and Sosdian et al. 2020 (Nat. Comm.) for C cycle in relationship to climate.

AC: Correct, we will provide more details on the estimated uncertainties used for error propagation, and will discuss the results in context of the new publications. Concerning Sosdian et al. (2020), the same comment came from referee 1 and we will include this contribution in the discussion.

Comparing CO2 records: e.g. Fig. 5: what drives the differences between different

d11B records for the target age-window? Section 4.1 needs some more discussion, with focus on how this new record could differ from previous d11B records. Could there be an upwelling signal at the study site driving those high CO2 estimates when they deviate from the other records? It could also be differences in the calibration used for d11B, or the assumptions for calculating carbonate system parameters. It may be wise to process the d11B records in the same manner, exclude the potential of any regional and variable CO2 disequilibrium, and then merge reliable d11B records into a single record with full propagation of uncertainties. If uncertainties are not propagated, and instead sensitivity runs are provided (i.e. d11Bsw), then better to display relative changes in pH/CO2 instead of absolute values. (Here it may be wise to remove the alkenone CO2 as they are not discussed enough beyond what is already available in the literature and thus do not contribute to the story).

AC: What the referee suggests here is what we applied for the boron-based reconstructions: We re-calculated pCO2 of the Foster et al. (2012) and Badger et al. (2013) data, using a TA of 2000 $\mu$mol/kg and a d11Bsw of 37.8 permil, as for our data. This was only mentioned in the Fig. 5 caption, so it was not adequately described in the main text, but will be done so in the revised manuscript. However, we will improve the data compilation and comparison by using exactly the same calibrations (although the differences between the ones used in the different studies are very small), as well as to account for the effect of d11Bsw on the calibrations (which might shift some reconstructions a bit). In addition, the effect of [Ca] and [Mg] of seawater on the dissociation constants will be included. Thanks for the suggestion to create a pCO2 "stack" from all reliable reconstructions. We will definitely try this, but at this point we expect it might not be too helpful since only few data exist with a too-low resolution. Concerning the possibility of a regional signal, the same question came from referee 1, and will be addressed in the revised manuscript.

Focusing on the d11B records available and this new one, it would be also beneficial to display not only CO2 but also pH evolution across the MMCT, and how different records

compare.

AC: As we re-calculated the boron-based pCO2 records, using identical boundary conditions (TA and d11BSw), we assume that also the pH curves will mimic the pCO2 curves, but we will certainly test this further.

Site setting: It is argued that the site is not affected by upwelling being north of the frontal systems in the Southern Ocean. However, can this be said with certainty for the middle Miocene? Is there any evidence for that?

AC: As mentioned earlier, we will address this question in the revised manuscript, which was also requested by reviewer 1.

Carbonate system calculations: The authors should consider the effect of Mg and Ca concentrations in seawater on carbonate system calculations (i.e. K1, K2, Ksp), such as in Hain et al. 2015; 2018 or Zeebe and Tyrrell 2019.

AC: Yes, good point. As mentioned earlier, this will be done for the revised manuscript.

Benthic-planktic pH records: Although the uncertainties are very large to make discernible conclusions about pH gradient values during the middle Miocene, it is interesting to further explore the dpH evolution and the surface-to-deep gradient evolution during the Miocene, and what drives this. If the benthic foraminiferal pH record is included, it should be discussed further.

AC: In our opinion, the resolution of the planktic and benthic pH isotope records in Figure S1 is too low to derive a meaningful interpretation concerning the temporal evolution of the surface to deep gradient. We would keep the figure in the supplement, but mention that the interpretation of the surface to deep gradient is limited by the low resolution of the data, especially the benthic curve is of lower resolution than the planktic curve.

Discussion on role of eccentricity and deep water ventilation: Here the section leaves us wanting more! It could benefit from some reorganization for clarity and flow, including recent studies such as those mentioned above.

AC: As this point is also raised by reviewer 1, the discussion will be re-organized and extended in the revised manuscript.

———————————————

---

## Author Response (AR1)

**Review No 1 (anonymous)**

Raitzsch et al present a new planktic boron isotope record across the mid Miocene Climate Transition, and interpret this in terms of changing CO2 and carbon cycle changes. The record enables deeper interrogation of potential climate and carbon cycle feedbacks during this important interval of global cooling. This interval also spans the end of the Monterey Excursion, which contains the well-known "carbon maxima" events. Having more CO2 data across this interval is very exciting, and thus the submission represents a substantial contribution worthy of publication in CP.

However, the narrative of the manuscript could be much improved – e.g. the conclusions are that a particular carbon cycle model is supported by the new records, but this model (or the mechanisms within it) are not referred to in the introduction. Much of the discussion hinges on assumptions of processes that have not been observed but are presented here as fact (e.g., that middle Miocene CO2 variations are caused by changes in shallow water carbonate production). The manuscript also fails to cite some key recent studies. However, I believe that if the narrative of the paper is improved and a few specific scientific questions are addressed then the robust interpretations of this fantastic dataset will be clearer and this will become a really exciting and useful contribution. Below I list the specific scientific questions I would like to see addressed, followed by more minor comments.

AC: We thank the referee for his constructive and thorough review of our manuscript. It was really helpful to look at our data from different perspectives discussed in more detail below.

*1. Is high eccentricity definitely associated with decreasing CO2? Would it be more robust to focus on the clear relationship between the d13C and CO2?*
The key finding presented in the abstract of the paper is that "long-term pCO2 variations between ~14.3 and 13.2 Ma were paced by 400 k.y. eccentricity cycles, with decreasing pCO2 at high eccentricity and vice versa." I struggle to see this relationship in Figure 4 or Figure 5 – i.e. an inverse correlation between the pCO2 record and the thin grey line. I'm not saying the pCO2 record doesn't contain a long eccentricity signal, but the temporal relationship with the eccentricity curve could perhaps be demonstrated more robustly. While I struggle to discern a negative correlation between the authors' pCO2 record and the long eccentricity cycle, the relationship that does seem convincing is the positive relationship between d13C and CO2 across CM6 (Figure 4 and described in section 3.1). Perhaps therefore a clearer and simpler approach would be to set out different mechanisms in the introduction with their respective d13C-CO2 relationships. E.g., Holbourn et al 2007 ascribes the eccentricity signal in the longer d13C record to EITHER increased productivity and burial of Corg (the classic Monterey hypothesis) which would result in a negative correlation between d13C and CO2, OR a monsoon-driven increase in shallow water carbonate deposition, removing alkalinity from the oceans and releasing CO2, producing a positive correlation between d13C and CO2, as observed here. This explanation would be particularly useful because the increase in monsoon intensity variability has different impacts on the carbon cycle, but these are not discussed in the introduction. For example, an increase in nutrient delivery to the oceans and increasing Corg burial tends to increase d13C and decrease CO2. But in the Ma et al 2011 model this is outweighed by the concomitant increase in carbonate burial and associated CO2 release. It would be useful if this current manuscript could comment on the robustness of the relative importance of these processes in the model. However, see also my comment below about the uniqueness of CM6 - what might be expected to happen if for example the climate transition led to a general increase in marine productivity superimposed on these orbital variations? Should we expect the d13C-CO2 relationship for CM6 to hold true for all CM events? This complication should be clearly addressed in the introduction also.

AC: We still think that both d13C and pCO2 are linked to eccentricity variations, but there is a phase lag of d13C and pCO2 to long eccentricity in the order of 50 k.y., which is possibly related to the slow response of weathering feedbacks to orbital forcing. This is now clearer from the text (l. 15-16; 222-225; 329-331; 403-404). Also we further emphasize the close relationship between d13C and pCO2 (l. 13-15; 221-222; 256-257; 326-327). However, we also slightly changed the title to not declare eccentricity as the only possible mechanism influencing our record. The introduction has been restructured to better explain the different hypotheses and the different d13C-pCO2 relationships associated with them (l. 35-51), also including the new study by Sosdian et al. (2020). In addition, we introduce the reader to the uniqueness of CM6 and the effect of Antarctic glaciation on the carbon cycle (l. 52-64), which also includes the recent study by Leutert et al. (2020) and Ma et al. (2018). Moreover, as the model of Ma et al. (2011) we refer to was designed for an ice-free world, we discuss the potential effect of Antarctic ice-sheet expansion on the carbon cycle in more detail (l. 358-372).

***2. What is the evidence for the increased shallow water carbonate deposition?***
As far as I know there is no direct evidence for eccentricity paced changes in shallower water carbonate burial. There are dissolution cycles in deep sea carbonates, but I am not aware of any study that rules out the influence of bottom water mass ventilation on these? I believe Holbourn et al 2007 favoured the monsoon hypothesis due to the 50 kyr lag between eccentricity and d13C. But this lag is not discussed in this manuscript, and no supporting evidence is given for the statements that shallow water carbonate burial changed at this time.

AC: We provide a number of references that deal with regional studies of increased carbonate deposition, but also mention that it is difficult to assess global trends from those (l. 342-348).

***3. Does the Site 1092 planktic d11B record global pCO2?***
This relationship between d13C and CO2 is key, and is independent of age model issues because the d13C has been recorded at each site and is a global signal. So it is interesting therefore that the Malta record shows a decrease in d11B-derived pCO2 associated with the onset of CM6 (increasing d13C) whereas the 1092 d11B-derived pCO2 shown here shows an increase. This raises the important question – is Site 1092 recording global atmospheric pCO2, or a more localised signal? A more local signal showing C storage in the high latitudes of the south Atlantic over the MMCT would also be very interesting of course. The change in the frontal positions is acknowledged in the text (Kuhnert et al 2009) but the change in stratification at this time is not mentioned. Paulsen (2005) (Bremen thesis) shows an increase in stratification at Site 1092 starting at 13.85 Ma (using the divergence of surface:deep planktonic foraminiferal d18O). Could this stratification have been associated with an increase in surface water [CO2]? This possibility should be addressed in the manuscript.

AC: We have added an entire new section ("4.1 Origin of Site 1092 $pCO_2$ signal") to tackle this question, discussing whether Site 1092 could have recorded a regional signal or not, with the conclusion that we cannot completely rule out local processes (l. 255-282).

***4. Is it appropriate to compare the records across CM6 with those of a model that does not consider the impacts of the MMCT itself?***
CM6 immediately follows the ice growth of the MMCT and is the largest by far of the CM events, making its interpretation more complex than the other CM events. It follows a major sea level fall, which would have affected the shelf:basin burial of carbonate, d13C and CO2 (e.g., Mckay et al 2016, Ma et al 2018). Further, the ice advance was likely associated with changes in ocean circulation and ventilation of bottom water masses (with implications for both d13C and CO2). Is it therefore

appropriate to make a straightforward comparison of the d13C and CO2 records to the Ma et al 2011 model without any consideration of these other processes? The Ma et al 2011 model was constructed to explain the long eccentricity signal in the d13C record throughout the Miocene Climatic Optimum, rather than examine CM6 specifically. It is solely forced by ETP, and does not include processes triggered by cryospheric thresholds in the climate system and resulting impacts. On the other hand, the Ma et al (2018) model suggests that a significant cause of the d13C increase at CM6 was the increased weathering resulting from the sea level fall. By saying the results here support the Ma et al 2011 model for CM6, where does that leave our understanding of the importance of shelf-basin carbonate deposition on global d13C signals?

AC: The reviewer correctly criticized that we refer to the model of Ma et al. (2011), which was designed for an ice-free world. Hence, we discuss the potential effect of Antarctic ice-sheet expansion superimposed on weathering on the carbon cycle in more detail (l. 358-372).

***5. How does the pCO2 record compare with the high-resolution B/Ca record of Sosdian et al 2020?***
It is odd that the manuscript does not compare the d11B-CO2 record with the high resolution B/Ca records published by Sosdian et al 2020. Those authors suggest that the increasing d13C of the CM events is associated with decreasing surface water DIC at Site 761. This is consistent with an increase in Corgburial:Carbonate burial which would predict a negative relationship between d13C and global CO2 across CM events, rather than the positive relationship observed across CM6 at Site 1092 here. Although, before a direct comparison can be made the origin of the d11B signal at Site 1092 needs to be thoroughly addressed.

AC: We have added a discussion of the new study by Sosdian et al. (2020) (l. 43-48; 397-313).

***6. What is the significance of the change in the Malta age model for the GSSP?***
The GSSP for the Langhian-Serravallian boundary is placed in the Malta section. There should be some comment about this in the age model section as the authors have changed the Malta age model.

AC: We have added a statement to this section, stating that that the revised age model is valid, although we did not explicitly place the boundary in there (l. 100-103).

*More minor comments:*

Line 45: *"However, most proxy records for the history of pCO2 across the MMCT are incomplete or at low resolution, thus prohibiting resolution of the CM events (Pagani et al., 1999; Kürschner et al., 2008; Foster et al., 2012; Ji et al., 2018; Sosdian et al., 2018; Super et al., 2018) and making it difficult to identify the mechanism responsible for the major step into the "icehouse" world."*
This statement is a bit disingenuous, these published records clearly show a significant pCO2 decrease just prior to the MMCT (see also Sosdian et al 2020). What is interesting of course is the higher resolution CO2 changes, and questions such as: what caused the CM events? how do the CM events depend on background climate state? Why is CM6 (immediately following the major ice growth) the largest of the CM events?

AC: We do not contradict existing studies showing a pCO2 that is lower after EAIS expansion than before. Actually we see something similar, but our record shows more details across CM events 5b and 6 (Fig. 6). We thoroughly discuss what could have caused the observed pCO2 and d13C evolution (l. 284-398) and summarize our conclusions in section 5 (l. 400-416).

Line 35: *"In a recent study, it was proposed that a more sluggish meridional Pacific Ocean overturning circulation, due to reduced deep-water formation in the Southern Ocean, enhanced the weathering of 13C-enriched shelf carbonates"*.
This is incorrect, the enhanced weathering in this model was ascribed to the sea level fall. This model (Ma et al 2018) is very different to the Ma et al 2011 model. The Ma et al 2018 model treats the MMCT as an "event" whereas the Ma et al 2011 uses continually evolving orbital parameters as forcing. This important point is not explained in the current manuscript.

AC: The summary of the Ma et al. (2018) study has been corrected (l. 60-64, 361-363).

I find it odd that pH and pCO2 are plotted as the same curve with the same uncertainty envelope in Figure 4. pCO2 clearly has additional uncertainties (e.g. estimate of TA, d11Bsw), and I think these should be better represented in Figures 4 and 5.

AC: This was also an issue raised by reviewer 2. We have re-calculated all pCO2 values, temperatures and associated uncertainties using a Monte Carlo approach, which propagates all specific uncertainties in B isotope and Mg/Ca measurements, calibration regressions, TA, Sal and d11B of seawater. Further, we account for the effect of Miocene [Mg,Ca] of seawater on the equilibrium constants and on Mg/Ca temperatures, as well as the effect of Miocene d11Bsw on the intercept of the applied B isotope calibration. The method part on this all has been edited, extended and restructured (l. 123-207). In addition, we created a new figure (S1) to show the effect of d11Bsw extremes on calculated pCO2.

Line 31: Need to also cite Foster et al 2012 as demonstrating climate-CO2 relationship across MMCT.

AC: Added (l. 33).

Careful with all references – e.g. Fig 3 caption "Badger et al 2015" should be "Badger et al 2013".

AC: Corrected (l. 716).

Line 168 – Fig 5 should be Fig 4.

AC: Corrected.

**Review No 2 (anonymous)**

GENERAL COMMENTS Obtaining more B isotope-pH and CO2 estimates for the middle Miocene climate transition is a long overdue goal, making this study very timely and of great importance. The authors provide a comprehensive view of CO2 evolution for this period and potential mechanisms overlying potential eccentricity driven variations. The paper would benefit from some re-organization and focus on clarity, incorporation of recent studies in the discussion and the data, and a more comprehensive propagation of uncertainties, beyond the sensitivity analyses performed for alkalinity and salinity.

AC: We thank the referee for his constructive and thorough review of our manuscript, particularly concerning the analytical and data handling part. We have addressed all questions and comments, which are listed in more detail below.

SPECIFIC COMMENTS Uncertainties: a) The analytical uncertainty reported is quite small compared to the uncertainty of replicate analyses. There may be differences between the use of IC vs. Faraday detectors in different studies and here. Nevertheless, the authors should provide more details here as well on how they calculate analytical uncertainty. For example, the consistency standard used to calculate long term precision should have been run at similar concentrations to those of samples, and the uncertainty of this should be larger at low B-levels. Additionally, the authors should provide more details on the B blank contribution (if any).

AC: Only looking at the control standards measured at 2-3 ppb such as in this study, the long-term reproducibility is not different from the of 0.3 ‰ (2*SD) we have reported in the previous version. What was not clear from the previous manuscript is that the control standard is normally run ~6 times or more within a session, also in order to condition and stabilize the system before measuring the samples. At the end of a session, the average of the measured control standards is calculated. The 2*SD of the averages between all sessions is ~0.3 ‰ (which we take as the minimum uncertainty), while it is ~0.6 ‰ when all single analyses are considered. We therefore added the information that the long-term reproducibility is calculated from "per-session" averages of a control standard (l. 120). We also added some brief information on the procedural blanks (l. 110-111).

b) Why is d11Bsw error systematic? If weathering is extremely pronounced, couldn't this cause variations in d11Bsw across this time window, even if the average residence time for B may be longer? Even if so, because of the non-linearity of the d11B-pH proxy, at different d11Bsw the dpH and thus dCO2 could differ. Some could, thus, argue the uncertainty in d11Bsw encompasses both uncertainty in absolute value across the MMCT but also potential variations across the window. The authors should provide at minimum two scenarios based on minimum and maximum d11Bsw estimates for this period.

AC: The uncertainty in d11Bsw is supposed to be systematic, even if weathering varied a lot, due to the large reservoir of the ocean vs rivers, and the B concentrations of those (ca. 4500 vs 16 ppb, respectively). However, as we re-calculated all pH, pCO2 and temperature values plus their associated uncertainties, which are propagated from specific uncertainties in input parameters, we also applied an uncertainty of 0.2‰ for d11B of seawater to account for potential variations in weathering. In addition, as the reviewer suggested, we added a new figure (S1) to demonstrate the effect of extreme d11Bsw values on calculated pCO2 values.

c) The level of details in Fig. 7 with all sensitivity analyses is very much appreciated. However, could the authors provide more explanation on how they estimate the Alk and Temp uncertainties? If they compare to literature or proxy estimates, shouldn't they use the maximum uncertainty reported (i.e. ± 2C, and ± 130 umol/kg Alk)? Comparing to other studies: The authors should discuss their results in light of two recent publications for the middle Miocene, Leutert et al. 2020 (Nat. Geo) for both their SST and dpH estimates, and Sosdian et al. 2020 (Nat. Comm.) for C cycle in relationship to climate.

AC: We provide more details on estimates of uncertainties and error propagation in the completely restructured method section 1.4 (l. 123-207). We also included discussions of the new studies by Sosdian et al. (2020) and Leutert et al. (2020) (l. 43-48; 58-60; 276-279; 289-292; 297-307).

Comparing CO2 records: e.g. Fig. 5: what drives the differences between different d11B records for the target age-window? Section 4.1 needs some more discussion, with focus on how this new record could differ from previous d11B records. Could there be an upwelling signal at the study site driving those high CO2 estimates when they deviate from the other records? It could also be differences in the calibration used for d11B, or the assumptions for calculating carbonate system parameters. It may be wise to process the d11B records in the same manner, exclude the potential of any regional and variable CO2 disequilibrium, and then merge reliable d11B records into a single record with full propagation of uncertainties. If uncertainties are not propagated, and instead sensitivity runs are provided (i.e. d11Bsw), then better to display relative changes in pH/CO2 instead of absolute values. (Here it may be wise to remove the alkenone CO2 as they are not discussed enough beyond what is already available in the literature and thus do not contribute to the story).

AC: We have added an entire new section ("4.1 Origin of Site 1092 $p$CO$_2$ signal") to tackle this question, discussing whether Site 1092 could have recorded a regional signal or not, with the conclusion that we cannot completely rule out local processes (l. 255-282). Moreover, we have improved our approach to better compare or record with other boron-based pCO2 reconstructions by applying exactly the same calculation procedure and using the same boundary conditions. However, we did not merge the records into a single one because of the few published data in rather low resolution. Further, there seem to be in part differences between the records, which would have skewed the fitted running average. We still show the alkenone records for comparison, but added a brief discussion why those are possibly not reliable at low pCO2 levels (l. 292-296).

Focusing on the d11B records available and this new one, it would be also beneficial to display not only CO2 but also pH evolution across the MMCT, and how different records compare.

AC: As we re-calculated the boron-based pCO2 records, using identical boundary conditions (TA and d11BSw), we do not show the pH curves in extra plots as they just mimic the pCO2 curves.

Site setting: It is argued that the site is not affected by upwelling being north of the frontal systems in the Southern Ocean. However, can this be said with certainty for the middle Miocene? Is there any evidence for that?

AC: As mentioned earlier, we have added an entire new section ("4.1 Origin of Site 1092 $p$CO$_2$ signal") to tackle this question, discussing whether Site 1092 could have recorded a regional signal or not, with the conclusion that we cannot completely rule out local processes (l. 255-282).

Carbonate system calculations: The authors should consider the effect of Mg and Ca concentrations in seawater on carbonate system calculations (i.e. K1, K2, Ksp), such as in Hain et al. 2015; 2018 or Zeebe and Tyrrell 2019.

AC: Thanks for this important hint. We have performed all calculations using the effect of [Mg,Ca] on the equilibrium constants. The procedure for this is explained in the new subsection "1.4.2 Equilibrium constants" (l. 142-147).

Benthic-planktic pH records: Although the uncertainties are very large to make discernible conclusions about pH gradient values during the middle Miocene, it is interesting to further explore the dpH evolution and the surface-to-deep gradient evolution during the Miocene, and what drives this. If the benthic foraminiferal pH record is included, it should be discussed further.

AC: The resolution of the planktic and benthic pH records in Figure S4 is too low and uncertainties too large to draw substantiated conclusions concerning the temporal evolution of the surface-to-deep gradient. This is now better explained (l. 216-220), but we could use the data to possibly exclude decreased vertical mixing at this site after glaciation (l. 274-276).

Discussion on role of eccentricity and deep water ventilation: Here the section leaves us wanting more! It could benefit from some reorganization for clarity and flow, including recent studies such as those mentioned above.

AC: The discussion has been much extended and restructured to enhance clarity and flow (l. 284-398).

---

## Author Response (AR2)

**Review No 1 (anonymous)**

I am pleased to see the authors address my comments so comprehensively (although I do not know why they assume I am male!). The discussion about the interpretation of the 1092 record as a global signal is very thoughtful. The whole paper is now much clearer as the authors have explained the nature of CM6 in the context of the ice sheet growth at the MMCT. The only change I now suggest is that "falling sea level" on line 45 should be "rising sea level."

AC: We thank the referee for his/her effort to review our manuscript again. And thanks for spotting this slip, we have changed "falling" to "rising" sea level (l. 45).

**Review No 2 (anonymous)**

Raitzsch et al. present us a relatively high resolution record of atmospheric CO2 from a Southern Ocean site (ODP Site 1092) for the middle Miocene climate transition. The manuscript, both data processing and interpretation, has been substantially improved, and it will be a great addition to the scientific toolkit on climate-carbon-ice sheet interactions during the Neogene and beyond.
I only have a few additional comments and suggestions, for which indicated line numbers refer to the marked version of the manuscript.

AC: We thank the referee for his/her effort to review our manuscript again. The replies to the additional comments are listed below.

1.1 Sampling strategy: please include the size fraction selected.

AC: We used the 250-315 µm size fraction, and added this information (l. 83-84).

Line 165: Please consider rephrasing as (or similar to): "By contrast, for T. trilobus d11B the T. sacculifer calibration is applied, with four available equations ( …..). The calibrations refer to different size fractions of T. sac and different analytical techniques…"
T. sacculifer d11B calibrations are dependent on the size fraction of the foraminifera shells (e.g. see summary in Foster et al. 2012). Comparing individual records with different calibrations without considering the size fraction of the foraminifera used might be misleading.

AC: Good point. The sentence has been rephrased accordingly (l. 152-155).

Lines 209-212: Mg/Ca is pH insensitive for T. sac (Gray and Evans 2019). Therefore the pH correction should not be applied to T. trilobus, but it is not clear if it is.

AC: For the sake of consistency, we applied the modified 'MgCaRB' model for the T. trilobus records, too. But, as the reviewer stated correctly, there is no discernible pH effect on Mg/Ca of this species. This is now clear in the text (l. 186-189).

Lines 261-262. The CO2 record has large uncertainties on the scale of ~ 50-170 ppm. How confidently could we interpret variations on the scale of 50-100 ppm? The ~100 ppm step change at 13.82 Ma is probably meaningful as it displays some consistency with the following CO2 estimates and the general cyclicity of the record, but the 50 ppm higher frequency variations remain a question. Relative CO2 changes must carry a smaller uncertainty, and an estimate of this would be very useful to demonstrate that such variations are indeed meaningful and interpretable.

AC: That is an interesting point raised by the reviewer, and indeed the uncertainty of pCO2 estimates is much smaller if we neglect the uncertainties of parameters that are unlikely to change on shorter timescales, i.e. those for the calibration equation and d11B of seawater. This is now addressed in the manuscript (l. 219-222).

Lines 355 and below:
At the risk of "wiggle matching", comparing the B/Ca record of Sosdian et al., 2020 with this CO2 record one might see some striking similarities. There is a long term decline in CO2, just as the B/Ca record displays a long term increase (and thus decline in surface water DIC). Within the overlapping period, there are two clear ~ 400ky larger cycles, each of which include a CM. The cycle that includes CM5b is characterised by both lower B/Ca (higher DIC) and higher CO2, thus both records appear consistent. The cycle that includes CM6 is intriguing because of the structure within the B/Ca record of Sosdian et al., 2020 in comparison to the d11B and B/Ca in Badger et al., 2013 and this CO2 record. If we ignore the point at ~13.8 Ma of the record in Badger et al., 2013, and focus on the period transitioning into CM6, B/Ca and d11B/CO2 follow the expected covariation so as higher d11B (higher pH and lower CO2) occurs when B/Ca is higher (lower DIC). But then, the expected minima in B/Ca at CM6, when compared to this CO2 record here, and the correlation of records within CM5b, collapses at the peak of the d13C positive excursion but recovers shortly after. Of course, the disagreement could be driven by issues on either of the two proxies, e.g. non-DIC effects on B/Ca, or the assumptions regarding the d11B-CO2 proxy, or regional effects on either of these two records, or a more global scale change. It would be great if both the similarities and the disagreements can be explored, and in that context identify what could make CM6 different to CM5b.

AC: We fully agree that a deeper comparison of our record with the B/Ca record of Sosdian et al. (2020) is beneficial, and have therefore extended section 4.2 (l. 314-320).

Line 422 and below:
Clearly CM6 is characterized by a more pronounced d13C change compared to all the other CM events. However, this is not something observed in this CO2 record. If anything, comparing the maximum and minimum one might say CM5b CO2 is more pronounced than CM6, since the maximum is ~equivalent but the minimum of CM5b CO2 is clearly higher, and the amplitude of change is either equivalent or larger for CM5b than CM6. This comparison gets further complicated by the argument of enhanced deep water formation during CM6 (which is plausible), which would likely place Site 1092 into a frontal system in the Southern Ocean, recording higher [CO2] than atmospheric CO2 (and possibly also contributing to the divergence in the records of B/Ca and CO2). What would drive the difference in amplitude of change between the d13C and CO2 records for CM6 in comparison to the other CM?

AC: While we agree that answering the questions about the exceptional nature of CM6 in comparison to the other CM events would be a great step forward in Miocene climate research, we think that, based on our data, the room for new speculations is quite limited. However, we added a concluding remark at the end of section 4.4 (l. 413-419) emphasizing again that Site 1092 may bear a regional signal that differs from global climate evolution, and that further research on this topic is required.

**Other changes**

In addition to the changes suggested by the reviewers, we found that in the supplemental Fig. S2 "G. trilobus" should be "T. trilobus". We have corrected the figure accordingly.